# Finite Continuum-Armed Bandits

Solenne Gaucher

Laboratoire de Mathématiques d'Orsay
Université Paris-Saclay,
91405, Orsay, France
solenne.gaucher@math.u-psud.fr

## Abstract

We consider a situation where an agent has $T$ ressources to be allocated to a larger number $N$ of actions. Each action can be completed at most once and results in a stochastic reward with unknown mean. The goal of the agent is to maximize her cumulative reward. Non trivial strategies are possible when side information on the actions is available, for example in the form of covariates. Focusing on a nonparametric setting, where the mean reward is an unknown function of a one-dimensional covariate, we propose an optimal strategy for this problem. Under natural assumptions on the reward function, we prove that the optimal regret scales as $O(T^{1/3})$ up to poly-logarithmic factors when the budget $T$ is proportional to the number of actions $N$. When $T$ becomes small compared to $N$, a smooth transition occurs. When the ratio $T/N$ decreases from a constant to $N^{-1/3}$, the regret increases progressively up to the $O(T^{1/2})$ rate encountered in continuum-armed bandits.

## 1 Introduction

### 1.1 Motivations

Stochastic multi-armed bandits have been extensively used to model online decision problems under uncertainty : at each time step, an agent must choose an action from a finite set, and receives a reward drawn i.i.d. from a distribution depending on the action she has selected. By choosing the same action over and over again, she can learn the distribution of the rewards for performing this action. The agent then faces a trade-off between collecting information on the mechanism generating the rewards, and taking the best action with regards to the information collected, so as to maximise her immediate reward.

In some real-life situations, the agent can complete each action at most once, and does not have enough resource to complete all of them. Her decisions can be rephrased in terms of allocating limited resources between many candidates. The agent cannot estimate the reward of an action by performing it several times, and must rely on additional information to construct her strategy. In many situations, covariates providing information on the actions are available to the agent. Then, the expected reward for taking an action can be modelled as a (regular) function of the corresponding covariate. Thus, similar actions give rise to similar rewards. This problem is motivated by the following examples.

- **Allocation of scarce resources.** The response of an individual to medical treatment can be inferred from contextual information describing this patient. When this treatment is expensive or short in supply, decision-makers aim at efficiently selecting recipients who will be treated, so as to maximise the number of beneficial interventions [Kleinberg et al., 2015]. During epidemic crises, lack of medical resources may force hospital staff to progressively identify patients that are more likely to recover based on indicators of their general health status, and prioritize them in the

resource allocation. Similar questions arise when determining college admission so as to optimize the number of successful students [Kleinberg et al., 2018], or allocating financial aid to individuals most likely to benefit from it.

- **Contextual advertisement with budget.** A common form of payment used in online advertisement is pay-per-impression: the advertiser pays a fixed fee each time an ad is displayed [Combes et al., 2015], and the budget of an advertising campaign determines the number of users who view the advertisement. It has been shown in [Agarwal et al., 2009] that click-through rates decrease steeply as users are exposed over and over to the same recommendation. Advertisers may therefore prefer to display their campaign to a new potential customer rather than to an already jaded one, so that each user will view the campaign at most once. Those users are often described by features including demographic information as well as previous online activities. Advertisers want to leverage this contextual information so as to focus on users that are more likely to click on the ad banner.

- **Pair matching.** Finding good matches between pairs of individuals is an ubiquitous problem. Each pair of individuals represents an action : the agent sequentially selects $T$ pairs, and receives a reward each time the pair selected corresponds to a good matching [Giraud et al., 2019]. In many settings, the agent has access to information describing either individuals or pairs of individuals. For example, online gaming sites may want to pair up players of similar level or complementary strength; dating applications may use information provided by the users to help them find a partner. Similarly, biologists studying protein-protein interaction networks will sequentially test pairs of proteins to discover possible interactions. Such experiments are however costly, difficult and time-consuming, and leveraging information describing those proteins can help researchers focus on pairs more likely to interact [Szilagyi et al., 2005].

In these settings, the decision maker can complete each action (i.e., select each internet user, patient, college candidate or pair of individuals) at most once; however by selecting an action, she learns about the expected rewards of similar actions. We model the dependence of the expected reward on the variable describing this action in a non-parametric fashion, and rephrase our problem by using terminology from the bandit literature.

**The Finite Continuum-Armed Bandit (F-CAB) problem :** An agent is presented with a set of $N$ arms described by covariates $\{a_1, a_2, ..., a_N\}$ in a continuous space $\mathcal{X}$ (the arm $i$ will henceforth be identified with its covariate $a_i$). The agent is given a budget $T$ to spend on those arms, where $T$ is typically a fraction $p$ of the number of available arms $N$. At each step $t \leq T$, the agent pulls an arm $\phi(t)$ among the arms that have not been pulled yet, and receives the corresponding reward $y_{\phi(t)} \in [0, 1]$. Conditionally on $\{a_1, a_2, ..., a_N\}$, the rewards $y_i$ are sampled independently from some distribution with mean $m(a_i)$, where $m : \mathcal{X} \to [0, 1]$ is the (unknown) mean reward function. The aim of the agent is to maximise the sum of the rewards she receives.

The F-CAB problem is closely related to the classical continuum-armed bandit problem. This problem, first introduced in [Kleinberg, 2004], extends multi-armed bandits to continuous sets of actions. At each step, the agent takes an action indexed by a point of her choosing in a continuous space $\mathcal{X}$. In order to maximise her gains, she must explore the space $\mathcal{X}$ so as to find and exploit one of the maximas of the mean reward function. The assumption that the agent can choose arbitrarily any action corresponding to any covariate, unrealistic in many real-life situations, is relaxed in the F-CAB model. Moreover in the F-CAB setting, the agent can pull each arm at most once. Thus she must endeavour to find and exploit a large set of good arms, as she cannot focus on a single arm corresponding to a maxima.

## 1.2 Related work

To the best of the author's knowledge, continuum-armed bandits without replacement have not been considered before. On the other hand, variants to the multi-armed bandit problem were proposed to relax the assumption that the agent can choose any action an infinite number of time.

In [Chakrabarti et al., 2009], the authors consider a multi-armed bandit problem with infinitely many arms, whose rewards are drawn i.i.d. from some known distribution. Each arm can only be pulled a finite number of times before it dies. Algorithms developed for this problem heavily rely on the knowledge of the distribution of the arms, and on the fact that an infinite number of good arms is always available to the player, both assumptions that are violated in our setting.

Closer to our problem is [Féraud and Urvoy, 2012] : the authors study the problem of scratch game, where each arm can be pulled a limited number of time before dying. They bound the weak regret, defined as the difference between $T \times m_{(\phi^*(T))}$ and the cumulative reward of the player, where $m_{(\phi^*(T))}$ is the expected reward of the $T$-th armed pulled by an oracle strategy. Since the reward of the arm pulled by this oracle strategy decreases at each step, its cumulative reward can be much larger than $T \times m_{(\phi^*(T))}$ (both can differ by a linear factor). Thus, the weak regret can be significantly lower than the classical regret, which we control in this paper.

Another related problem is that of budgeted bandits with different budgets for each arm : the decision maker faces a multi-armed bandit problem with constraints on the number of pull of each arm. This problem is studied in [Agarwal et al., 2009]: the authors assume that the number of arms is fixed, and that the budget of each arm increases proportionally to the number of steps $T$. They provide numerical simulations as well as asymptotic theoretical bounds on the regret of their algorithm. More precisely, they show that in the limit $T \to \infty$, all optimal arms but one have died before time $T$ : thus, when the budget of each arm and the total number of pulls $T$ are sufficiently large, the problem reduces to a classical multi-armed bandit. By contrast, in the F-CAB setting we can pull each arm at most once and do not attain this regime. Our technics of proof require therefore more involved, non-asymptotic regret bounds.

## 1.3 Contribution and outline

In this paper, we present a new model for finite continuum-armed bandit motivated by real-world applications. In this resource allocation problem, each action is described by a continuous covariate, and can be taken at most once. After some preliminary discussions, we restrict our attention to one-dimensionnal covariates and introduce further assumptions on the distribution of the covariates $a_i$ and on the mean payoff function $m$ in Section 2. In Section 3, we present an algorithm for this problem, and establish a non-asymptotic upper-bound on the regret of this algorithm. More precisely, we prove that when the budget $T$ is a fixed proportion of the number of arms, with high probability, $R_T = O(T^{1/3} \log(T)^{4/3})$. This rate is faster than all regret rates achievable in the classical continuum armed bandit under similar assumptions on the mean reward function. Indeed, the authors of [Auer et al., 2007] show that regret for the classical continuum-armed bandits problem is typically of order $O(T^{1/2} \log(T))$. On the other hand, we show that when the budget $T$ becomes small compared to the number of arms $N$, the regret rate smoothly increases. In the limit where the ratio $T/N$ decreases to $N^{-1/3} \log(N)^{2/3}$, the regret increases progressively up to the $O(T^{1/2} \log(T))$ rate encountered in classical continuum-armed bandit problems. Moreover, we derive matching lower bounds on the regret, showing that our rate is sharp up to a poly-logarithmic factor. Extensions of our methods to multi-dimensional covariates are discussed in Section 5 and detailed in the Appendix. We provide high level ideas behind those results throughout the paper but defer all proofs to the Appendix.

## 2 Problem set-up

### 2.1 Preliminary discussion

In the F-CAB problem, each arm can be pulled at most once, and exploration is made possible by the existence of covariates describing the arms. This framework is related to the classical Continuum-Armed Bandit problem, which we recall here.

**The Continuum-Armed Bandit (CAB) problem:** At each step $t$, an agent selects any covariate $a_t \in \mathcal{X}$, pulls an arm indexed by this covariate and receives the corresponding reward $y_t \in [0, 1]$. Here again, the rewards for pulling an arm $a \in [0, 1]$ are drawn i.i.d. conditionally on $a$ from some distribution with mean $m(a)$. The agent aims at maximising her cumulative reward.

By contrast to the CAB setting, where the agent is free to choose any covariate in $\mathcal{X}$, in the F-CAB setting she must restrict her choice to the ever diminishing set of available arms. The usual trade-off between exploration and exploitation breaks down, as the agent can pull but a finite number of arms in any region considered as optimal. Once those arms have been pulled, all effort spent on identifying this optimal region may become useless. On the contrary, in the CAB setting the agent may pull arms in a region identified as optimal indefinitely. For this reason, strategies lead to lower cumulative reward in the F-CAB setting than that in the less constrained CAB setting.

Nonetheless, this does not imply that F-CAB problems are more difficult than CAB ones in terms of regret. The difficulty of a problem is often defined, in a minimax sense, as the performance of the best algorithm on a worst problem instance. In bandit problems, the performance of a strategy $\phi$ is often characterised as the difference between its expected cumulative reward, and that of an agent knowing in hindsight the expected rewards of the different arms. At each step $t = 1, ..., T$, this oracle agent pulls greedily the arm $\phi^*(t)$, where $\phi^*$ denote a permutation of $\{1, ..., N\}$ such that $m(a_{\phi^*(1)}) \geq m(a_{\phi^*(2)}) \geq ... \geq m(a_{\phi^*(N)})$. Note that this agent receives an expected cumulative reward of $\sum_{t \leq T} m(a_{\phi^*(t)})$, which is lower than $T \times \max_a m(a)$. Thus the regret, defined as the difference between the cumulative reward of $\phi^*$ and that of our strategy, is given by

$$R_T = \sum_{1 \leq t \leq T} m(a_{\phi^*(t)}) - \sum_{1 \leq t \leq T} m(a_{\phi(t)}).$$

The difficulty of the F-CAB problem is governed by the ratio $p = T/N$. In the limit $p \to 1$, the problem becomes trivial as any strategy must pull all arms, and all cumulative rewards are equal. Opposite to this case, in the limit $p \to 0$, choosing $a_{\phi(t)}$ from the large set of remaining arms becomes less and less restrictive, and we expect the problem to become more and more similar to a CAB. To highlight this phenomenon, we derive upper and lower bounds on the regret that explicitly depend on $p$. We show that when $p \in (0, 1)$ is a fixed constant, i.e. when the budget is proportional to the number of arms, lower regret rates can be achieved for the F-CAB problem than for the CAB problem. To the best of the author's knowledge, it is the first time that this somewhat counter-intuitive phenomenon is observed; however it is consistent with previous observations on rotting bandits [Levine et al., 2017], in which the expected reward for pulling an arm decreases every time this arm is selected. Like in the F-CAB model, in rotting bandits the oracle agent receives ever decreasing rewards. The authors of [Seznec et al., 2019] show that this problem is no harder than the classical multi-armed bandit : although the cumulative rewards are lower than those in the classical multi-armed bandit setting, it does not imply that strategies should suffer greater regrets. This phenomenon is all the more striking in the F-CAB setting, as we show that strategies can in fact achieve lower regrets. Finally, we verify that when $p \to 0$, the regret rate increases. In the limit where $p = N^{-1/3} \log(N)^{2/3}$, the problem becomes similar to a CAB and the regret rate increases up to the rate encountered in this setting.

## 2.2 Assumptions on the covariates and the rewards

While in general the covariates $a_i$ could be multivariate, we restrict our attention to the one-dimensional case, and assume that $\mathcal{X} = [0, 1]$. The multivariate case is discussed and analysed in Section 5 and in the Appendix. Focusing on the one-dimensional case allows us to highlight the main novelties of this problem by avoiding cumbersome details. We make the following assumption on the distribution of the arms.

**Assumption 1.** *For $i = 1, ..., N$, $a_i \overset{i.i.d.}{\sim} \mathcal{U}([0, 1])$.*

By contrast to the CAB setting, where one aims at finding and pulling arms with rewards close to the maxima of $m$, in a F-CAB setting the agent aims at finding and pulling the $T$ best arms : the difficulty of the problem thus depends on the behaviour of $m$ around the reward of the $T$-th best arm $m(a_{\phi^*(T)})$. Under Assumption 1, we note that $\mathbb{E}[m(a_{\phi^*(T)})] = M$, where $M$ is defined as

$$M = \min \{A : \lambda(\{x : m(x) \geq A\}) < p\}$$

and $\lambda$ is the Lebesgue measure. In words, we aim at identifying and exploiting arms with expected rewards above the threshold $M$. We therefore say that an arm $a_i$ is optimal if $m(a_i) \geq M$, and that it is otherwise sub-optimal. Moreover, we say that an arm $a_i$ is sub-optimal (respectively optimal) by a gap $\Delta$ if $0 \leq M - m(a_i) \leq \Delta$ (respectively $0 \leq m(a_i) - M \leq \Delta$).

We make the following assumptions on the mean reward function. First, note that if $m$ varies sharply, the problem becomes much more difficult as we cannot infer the value of $m$ at a point based on rewards obtained from neighbouring arms. In fact, if $m$ presents sharp peaks located at the $T$ optimal arms, any reasonable strategy must suffer a linear regret. In order to control the fluctuations of $m$, we assume that it is weakly Lipschitz continuous around the threshold $M$.

**Assumption 2** (Weak Lipschitz condition). *There exists $L > 0$ such that, for all $(x, y) \in [0, 1]^2$,*

$$|m(x) - m(y)| \leq \max\{|M - m(x)|, L|x - y|\}. \tag{1}$$

Assumption 2 is closely related to Assumption A2 in [Bubeck et al., 2011]. It requires that the mean reward function $m$ is $L$-Lipschitz at any point $x'$ such that $m(x') = M$: indeed, in this case the condition states that for any $y$, $|m(x) - m(y)| \leq L|x - y|$. On the other hand, $m$ may fluctuate more strongly around any point $x$ whose expected reward is far from the threshold $M$.

Bandit problems become more difficult when many arms are slightly sub-optimal. Similarly, the F-CAB problem becomes more difficult if there are many arms with rewards slightly above or under the threshold $M$, since it is hard to classify those arms respectively as optimal and sub-optimal. This difficulty is captured by the measure of points with expected rewards close to $M$.

**Assumption 3** (Margin condition). *There exists $Q > 0$ such that for all $\epsilon \in (0, 1)$,*

$$\lambda\left(\{x : |M - m(x)| \leq \epsilon\}\right) \leq Q\epsilon. \tag{2}$$

In the classical CAB setting, lower bounds on the regret are of the order $O(T^{1/2})$ under similar margin assumptions, and they become $O(T^{2/3})$ when these margin assumptions are not satisfied. In the F-CAB, Assumption 3 allow us to improve regret bounds up to $O(T^{1/3}p^{-1/3})$. It is altogether not too restrictive, as it is verified if $m$ has finitely many points $x$ such that $m(x) = M$, and has non vanishing first derivatives at those points. Note that if the margin assumption and the weak Lipschitz assumption hold simultaneously for some $L, Q > 0$, we must have $QL \geq 1$.

## 3 UCBF : Upper Confidence Bound algorithm for Finite continuum-armed bandits

### 3.1 Algorithm

We now describe our strategy, the Upper Confidence Bound for Finite continuum-armed bandits (UCBF). It is inspired from the algorithm UCBC introduced in [Auer et al., 2007] for CAB.

---

**Algorithm 1** Upper Confidence Bound for Finite continuum-armed bandits (UCBF)

---

**Parameters:** $K, \delta$

**Initialisation:** Divide $[0, 1]$ into $K$ intervals $I_k$ with $I_k = [\frac{k-1}{K}, \frac{k}{K})$ for $k \in \{1, ..., K - 1\}$ and $I_K = [\frac{K-1}{K}, 1]$. Let $N_k = \sum_{1 \leq i \leq N} \mathbb{1}\{a_i \in I_k\}$ be the number of arms in the interval $I_k$. Define the set of intervals alive as the set of intervals $I_k$ such that $N_k \geq 2$. Pull an arm uniformly at random in each interval alive.

**for** $t = K + 1, ..., T$ **do**

    − Select an interval $I_k$ that maximizes $\widehat{m}_k(n_k(t-1)) + \sqrt{\frac{\log(T/\delta)}{2n_k(t-1)}}$ among the set of alive intervals, where $n_k(t-1)$ is the number of arms pulled from $I_k$ by the algorithm before time $t$, and $\widehat{m}_k(n_k(t-1))$ is the average reward obtained from those $n_k(t-1)$ samples.

    − Pull an arm selected uniformly at random among the arms in $I_k$. Remove this arm from $I_k$. If $I_k$ is empty, remove $I_k$ from the set of alive intervals.

**end for**

---

In order to bound the regret of UCBF, we show that it can be decomposed into the sum of a discretization term and of the cost of learning on a finite multi-armed bandit. First, we discuss the optimal number of intervals $K$. In a second time, we present new arguments for bounding more tightly the discretization error. Then, we show that by contrast to the classical CAB, the contribution of slightly sub-optimal arms to the regret is much more limited in F-CAB problems, before obtaining a high-probability bound on the regret of our algorithm.

By dividing the continuous space of covariates into intervals, we approximate the $\mathcal{X}$-armed setting with a finite multi-armed bandit problem, which we define bellow.

**The Finite Multi-armed Bandit (F-MAB) :** An agent is given a budget $T$ and a set of $K$ arms. At each step, the agent pulls an arm $k_t$ and receives a reward $y_t$ sampled independently with mean $m_{k_t}$. Each arm $k \in \{1, ..., K\}$ can only be pulled a finite number of time, denoted $N_k$, before it dies. The agent aims at maximising the sum of her rewards.

The approximation of the $N_k$ arms in an interval $I_k$ as a single arm that can be pulled $N_k$ times is done at the price of a discretization error, as we are now forced to treat all arms in the same interval

equally, regardless of possible differences of rewards within an interval. The choice of the number of intervals $K$ determines both the cost of this approximation, and the difficulty of the F-MAB problem. To analyse the dependence of those quantities on $K$, we introduce the strategy of an oracle agent facing the corresponding F-MAB problem (i.e., of an agent knowing in hindsight the expected mean rewards $m_k = \int_{I_k} m(a)da$ for pulling an arm in any interval $I_k$, and treating all arms in the same interval equally). We denote this strategy by $\phi^d$. Assume, for the sake of simplicity, that the intervals $I_1, ..., I_K$ have been reordered by decreasing mean reward, and that there exists $f \in \{1, ..., K\}$ such that $T = N_1 + ... + N_f$. Then, $\phi^d$ pulls all arms in the intervals $I_1$ up to $I_f$.

We can conveniently rewrite the regret $R_T$ as the sum of the regret of $\phi^d$, and of the difference between the cumulative rewards of $\phi^d$ and that of the strategy $\phi$ :

$$ R_T = \underbrace{\sum_{1 \leq t \leq T} m\left(a_{\phi^*(t)}\right) - \sum_{1 \leq t \leq T} m\left(a_{\phi^d(t)}\right)}_{R_T^{(d)}} + \underbrace{\sum_{1 \leq t \leq T} m\left(a_{\phi^d(t)}\right) - \sum_{1 \leq t \leq T} m\left(a_{\phi(t)}\right)}_{R_T^{(FMAB)}}. \qquad (3) $$

The regret $R_T^{(d)}$ is the regret suffered by an agent with hindsight knowledge of the expected mean rewards for the different intervals. It can be viewed as the discretization error. The additional regret $R_T^{(FMAB)}$ corresponds to the cost of learning in a F-MAB setting. All arms in an interval $I_k$ have a reward close to $m_k$, so by definition of $\phi^d$

$$ R_T^{(FMAB)} \approx \sum_{k \leq f} (N_k - n_k(T))m_k - \sum_{k > f} n_k(T)m_k. \qquad (4) $$

where we recall that $N_k$ denotes the number of arms belonging to interval $I_k$, and $n_k(T)$ denotes the number of arms pulled in this interval by UCBF at time $T$.

Choosing the number of intervals thus yields the following tradeoff : a low value of $K$ implies an easier F-MAB problem and a low value of $R_T^{(FMAB)}$, while a high value of $K$ allows for reduction of the discretization error. In finite bandits, exploration is limited : indeed, when increasing the number of intervals in a F-CAB setting, we simultaneously reduce the number of arms in each interval, and we may become unable to differentiate the mean rewards of two intervals close to the threshold $M$. Under the weak Lipschitz assumption, gaps between the rewards of two adjacent intervals are of the order $1/K$. Classical results indicate that $K^2$ pulls are needed to differentiate the mean rewards of those intervals. On the other hand, under Assumption 1, the number of arms in each interval is of the order $N/K$. Thus, choosing $K$ larger than $N^{1/3}$ will only increase the difficulty of the multi-armed problem, without reducing the discretization error (since $K^2 \geq N/K$ when $K \geq N^{1/3}$).

## 3.2 Bounding the discretization error

Equation (3) indicates that the regret can be decomposed as the sum of a discretization error and of the regret on the corresponding multi-armed bandit. In order to bound this discretization error, usual methods from continuum-armed bandits rely on bounding the difference between the expected reward of an arm and that of its interval by $L/K$. Thus, at each step, an algorithm knowing only the best interval may suffer a regret of the order $O(1/K)$, and the difference between the cumulative rewards of $\phi^d$ and $\phi^*$ is of the order $O(T/K)$. This argument yields sub-optimal bounds in F-CAB problems: indeed, the majority of the terms appearing in $R_T^{(d)}$ are zero, as $\phi^*$ and $\phi^{(d)}$ mostly select the same arms.

To obtain a sharper bound on the discretization error $R_T^{(d)}$, we analyse more carefully the difference between those strategies. More precisely, we use concentrations arguments to show that under Assumption 1, $m(a_{\phi^*(T)})$ and $m_f$ are close to $M$. This result implies that under the weak Lipschitz assumption, for any pair of arms $(a_i, a_j)$ respectively selected by $\phi^*$ but not by $\phi^d$ and vice versa, $m(a_i) - m(a_j) = O(L/K)$. Finally, the margin assumption allows us to bound the number of those pairs, thus proving the following Lemma.

**Lemma 1.** *Assume that $K \leq N^{2/3}$ and $K > p^{-1} \vee (1-p)^{-1}$. Under Assumptions 1, 2 and 3, there exists a constant $C_{L,Q}$ depending on $L$ and $Q$ such that with probability larger than $1 - 6e^{-2N/K^2} - 2e^{-N^{1/3}/3}$,*

$$R_T^{(d)} \leq C_{L,Q} \frac{T}{pK^2}.$$

We underline that this discretization error is lower than the unavoidable error of order $O(T/K)$, encountered in classical CAB settings.

### 3.3 Upper bound on the regret of UCBF

Before stating our result, we bound the regret due to slightly sub-optimal arms. It is known that in the classical CAB model, slightly sub-optimal arms contribute strongly to the regret, as any agent needs at least $O(\Delta^{-2})$ pulls to detect an interval sub-optimal by a gap $\Delta$. When $\Delta$ is smaller than $\sqrt{1/T}$, the agent spends a budget proportional to $T$ to test whether this interval is optimal or not, which leads to regret of the order $O(\Delta T)$. By contrast, in a F-CAB setting, pulling arms from an interval sub-optimal by a gap $\Delta$ until it dies, contributes to the regret by a factor at most $\Delta N/K$. Under Assumptions 1, 2 and 3, the number of intervals with mean rewards sub-optimal by a gap smaller than $\Delta$ is $O(K\Delta)$. Thus, we are prevented from mistakenly selecting those slightly sub-optimal intervals too many times. This is summarised in the following remark.

**Remark 1.** *Under hypothesis 1, 2 and 3, intervals sub-optimal by a gap $\Delta$ contribute to the regret by a factor at most $O(\Delta^2 T/p)$.*

Remark 1 along with Lemma 1 help us to bound with high probability the regret of Algorithm UCBF for any mean payoff function $m$ satisfying Assumptions 2 and 3, for the choice $K = \lfloor N^{1/3} \log(N)^{-2/3} \rfloor$ and $\delta = N^{-4/3}$. The proof of Theorem 1 is deferred to the Appendix.

**Theorem 1.** *Assume that $\lfloor N^{1/3} \log(N)^{-2/3} \rfloor > p^{-1} \vee (1-p)^{-1}$. Under Assumption 1, 2 and 3 there exists a constant $C_{L,Q}$ depending only on $L$ and $Q$ such that for the choice $K = \lfloor N^{1/3} \log(N)^{-2/3} \rfloor$ and $\delta = N^{-4/3}$,*

$$R_T \leq C_{L,Q} (T/p)^{1/3} \log(T/p)^{4/3}$$

*with probability at least $1 - 12(N^{-1} \vee e^{-N^{-1/3}/3})$.*

*Sketch of Proof.* We use Lemma 1 to bound the discretization error $R_T^{(d)}$. The decomposition in Equation (3) shows that it is enough to bound $R_T^{(FMAB)}$. Recall that $\phi^d$ pulls all arms in the intervals $I_1, I_2$, up to $I_f$, while UCBF pulls $n_k(T)$ arms in all intervals $I_k$. Using Equation (4), we find that

$$R_T^{(FMAB)} \approx \sum_{k \leq f}(N_k - n_k(T))(m_k - M) + \sum_{k > f} n_k(T)(M - m_k)$$

where we have used that $\sum_{k \leq f} N_k = T = \sum_{k \leq K} n_k(T)$, which in turns implies $\sum_{k \leq f} N_k - n_k(T) = \sum_{k > f} n_k(T)$.

On the one hand, $R_{subopt} = \sum_{k > f} n_k(T)(M - m_k)$ corresponds to the regret of pulling arms in sub-optimal intervals. We use Remark 1 to bound the contribution of intervals sub-optimal by a gap $O(1/K)$ by a factor of the order $O(T/(pK^2))$. Classical bandit technics allow to bound the contribution of the remaining sub-optimal intervals : under Assumptions 1–3, they contribute to the regret by a term $O(K \log(T) \log(K))$. Thus, for the choice $K = N^{1/3} \log(N)^{-2/3}$, we can show that $R_{subopt} = O((T/p)^{1/3} \log(T/p)^{4/3})$.

On the other hand, the term $R_{opt} = \sum_{k \leq f}(N_k - n_k(T))(m_k - M)$ is specific to finite bandit problems. The following argument shows that UCBF kills the majority of optimal intervals, and that optimal intervals $I_k$ alive at time $T$ are such that $f - k$ is bounded by a constant.

Let $I_k$ be an interval still alive at time $T$ such that $m_k > M$. Then the interval $I_k$ was alive at every round, and any interval selected by $\phi$ must have appeared as a better candidate than $I_k$. Using the definition of UCBF and Assumptions 3, we can show that the number of arms pulled from intervals with mean reward lower than $m_k$ is bounded by a term $O(N/K + K^2 \log(T))$.

Since $T = N_1 + ... + N_f$ arms are pulled in total, the number of arms pulled from intervals with mean reward lower than $m_k$ is at least $T - (N_1 + ... + N_k) = N_{k+1} + ... + N_f \approx (f-k)N/K$. Therefore, no interval $I_k$ such that $(f-k)N/K \geq O(N/K + K^2 \log(T))$ can be alive at time $T$. For the choice of $K$ described above, $(1 + K^3 \log(T)/N)$ is upper bounded by a constant. Thus, there exists a constant $C > 0$ such that for all $k \leq f - C$, all intervals $I_k$ have died before time $T$. We note that the number of arms in any interval is of the order $N/K$, so $R_{opt} = \sum_{f-C \leq k \leq f} (N_k - n_k(T))(m_k - M) \leq C(m_{(f-C)} - M)N/K$. To conclude, we use Assumption 2 to show that $m_{(f-C)} - M = O(CL/K)$, and find that $R_{opt} = O(N/K^2) = O(T/(pK^2))$. □

Under Assumptions similar to 2 and 3, [Auer et al., 2007] show that the regret of UCBC in CAB problems is $O(\sqrt{T} \log(T))$ for the optimal choice $K = \sqrt{T}/\log(T)$. By contrast, in the F-CAB problem, Theorem 1 indicates that when $p$ is a fixed constant, i.e. when the number of arms is proportional to the budget, the optimal choice for $K$ is of the order $T^{1/3} \log(T)^{-2/3}$ and the regret scales as $O(T^{1/3} \log(T)^{4/3})$. In this regime, regrets lower than that in CAB settings are thus achievable. As $N \to \infty$ and $p \to 0$, both the regret and the optimal number of intervals increase. To highlight this phenomenon, we consider regimes where $T = 0.5N^\alpha$ for some $\alpha \in [0, 1]$ (the choice $T \leq 0.5N$ reflects the fact that we are interested in settings where $T$ may be small compared to $N$, and is arbitrary). Theorem 1 directly implies the following Corollary.

**Corollary 1.** *Assume that* $T = 0.5N^\alpha$ *for some* $\alpha \in (2/3 + \epsilon_N, 1]$, *where we define* $\epsilon_N = \left(\frac{2}{3} \log\log(N) + \log(2)\right) / \log(N)$. *Then, for the choice* $\delta = N^{-4/3}$ *and* $K = \lfloor \alpha^{2/3} (2T)^{1/(3\alpha)} \log(2T)^{-2/3} \rfloor$, *with probability at least* $1 - 12(N^{-1} \vee e^{-N^{-1/3}/3})$,

$$R_T \leq C_{Q,L} T^{1/(3\alpha)} \log(T)^{4/3}$$

*for some constant* $C_{Q,L}$ *depending on Q and L.*

Corollary 1 indicates that as $\alpha$ decreases, the regret increases progressively from a F-CAB regime to a CAB regime. When the budget is a fixed proportion of the number of arms, the regret scales as $O(T^{1/3} \log(T)^{4/3})$ for the optimal number of intervals $K$ of the order $T^{1/3} \log(T)^{1/2}$. As $p$ decreases and $\alpha \in (2/3 + \epsilon_N, 1]$, the regret increases as $O(T^{1/(3\alpha)} \log(T)^{4/3})$ for $K$ of the order $T^{1/(3\alpha)} \log(T)^{-2/3}$. In the limit $\alpha \to 2/3 + \epsilon_n$, the regret rate becomes $R_T = O(\sqrt{T} \log(T))$ for the optimal number of intervals of the order $\sqrt{T}/\log(T)$, which corresponds to their respective values in the CAB setting.

To understand why $\alpha = 2/3 + \epsilon_N$ corresponds to a transition from a F-CAB to a CAB setting, note that $\alpha = 2/3 + \epsilon_N$ implies $T = N/K$ : in other words, the budget becomes of the order of the number of arms per interval. Thus, when $\alpha > 2/3 + \epsilon_N$, the oracle strategy exhausts all arms in the best interval, and it must select arms in intervals with lower mean rewards. In this regime, we see that the finiteness of the arms is indeed a constraining issue. On the contrary, if $\alpha \leq 2/3 + \epsilon_N$, no interval is ever exhausted. The oracle strategy only selects arms from the interval with highest mean reward, and our problem becomes similar to a CAB problem. Finally, we underline that when $\alpha \leq 2/3 + \epsilon_N$ the analysis becomes much simpler. Indeed, results can be directly inferred from [Auer et al., 2007] by noticing that no interval is ever exhausted, and that Algorithm UCBF is therefore a variant of Algorithm UCBC. In this case, the optimal choice for the number of intervals remains $K = \sqrt{T}/\log(T)$, and yields a regret bound $R_T = O(\sqrt{T} \log(T))$.

## 4   A lower bound

A careful analysis of the proof of Theorem 1 reveals that all intervals with mean reward larger than $M$ plus a gap $O(L/K)$ have died before time $T$. On the other hand, all intervals with mean rewards lower than $M$ minus a gap $O(L/K)$ have been selected but a logarithmic number of times. In other words, the algorithm UCBF is able to identify the set corresponding to the best $p$ fraction of the rewards, and it is only mistaken on a subset of measure $O(1/K)$ corresponding to arms $a_i$ such that $|m(a_i) - M| = O(1/K)$. We use this remark to derive a lower bound on the regret of any strategy for mean payoff function $m$ in the set $\mathcal{F}_{p,L,Q}$ defined bellow.

**Definition 1.** *For* $p \in (0, 1)$, $L > 0$ *and* $Q > 0$, *we denote by* $\mathcal{F}_{p,L,Q}$ *the set of functions* $m : [0, 1] \to [0, 1]$ *that satisfy Equations* (1) *and* (2).

To obtain our lower bound, we construct two functions $m_1$ and $m_2$ that are identical but on two intervals, each one of length $N^{-1/3}$. On those intervals, $m_1$ and $m_2$ are close to the threshold $M$ separating rewards of the fraction $p$ of the best arms from the rewards of the remaining arms. One of these intervals corresponds to arms with reward above $M$ under the payoff function $m_1$ : more precisely, on this interval $m_1$ increases linearly from $M$ to $M + 0.5LN^{-1/3}$, and decreases back to $M$. On this interval, $m_2$ decreases linearly from $M$ to $M - 0.5LN^{-1/3}$, and increases back to $M$. We define similarly $m_1$ and $m_2$ on the second interval by exchanging their roles, and choose the value of $m_1$ and $m_2$ outside of those intervals so as to ensure that both functions belong to the set $\mathcal{F}_{L,Q}$ for some $Q$ large enough.

Now, any reasonable strategy pulls arms in both intervals until it is able to differentiate the two mean reward functions, or equivalently until it is able to determine which interval contains optimal arms. As the average payments of those two intervals differ by $\Omega(N^{-1/3})$, this strategy must pull $\Omega(N^{2/3})$ arms in both intervals. This is possible since there are $N^{2/3}$ arms in each interval. Since arms in one of those intervals are sub-optimal by a gap of the order $N^{-1/3}$, this strategy suffers a regret $\Omega(N^{1/3})$.

In order to formalise this result, we stress the dependence of the regret on the strategy $\phi$ and the mean reward function $m$ by denoting it $R_T^\phi(m)$. Our results are proved for reward $y$ that are Bernoulli random variables (note that this is a special case of the F-CAB problem).

**Assumption 4.** *For $i \in \{1, ..., N\}$, $y_i \sim Bernoulli(m(a_i))$.*

In order to simplify the exposition of our results, we assume that the arms $a_i$ are deterministic.

**Assumption 5.** *For $i \in \{1, ..., N\}$, $a_i = \frac{i}{N}$.*

**Theorem 2.** *For all $p \in (0, 1)$, all $L > 0$, all $Q > (6/L \vee 12)$, there exists a constant $C_L$ depending on $L$ such that under Assumptions 5 and 4, for all $N \geq C_L(p^{-3} \vee (1 - p)^{-3})$,*

$$\inf_\phi \sup_{m \in \mathcal{F}_{p,Q,L}} \mathbb{P}\left(R_T^\phi(m) \geq 0.01 T^{1/3} p^{-1/3}\right) \geq 0.1.$$

Theorem 2 shows that the bound on the regret of UCBF obtained in Theorem 1 is minimax optimal up to a polylogarithmic factor. The proof of Theorem 1 is deferred to the Appendix. Again, we stress the dependence of this regret bound on $T$ by considering regimes where $T = 0.5N^\alpha$. The following Corollary follows directly from Theorem 2.

**Corollary 2.** *For all $L > 0$, $Q > (6/L \vee 12)$, there exists a constant $C_L$ depending on $L$ such that such that for all $N > \exp(3C_L)$ and all $T$ such that $T = 0.5N^\alpha$ for some $\alpha \in (2/3 + C_L/\log(N), 1]$, under Assumptions 5 and 4,*

$$\inf_\phi \sup_{m \in \mathcal{F}_{0.5N^{\alpha-1},Q,L}} \mathbb{P}\left(R_T^\phi(m) \geq 0.01 T^{1/(3\alpha)}\right) \geq 0.1.$$

## 5  Discussion

We have introduced a new model for budget allocation with short supply when each action can be taken at most once, and side information is available on those actions. We have shown that, when covariates describing those actions are uniformly distributed in $[0, 1]$, the expected reward function $m$ satisfies Assumption 2 and 3, and the budget is proportional to the number of arms, then the optimal choice of number of intervals $K$ is of the order $T^{1/3} \log(T)^{-2/3}$, and the regret is $O(T^{1/3} \log(T)^{4/3})$. Our lower bound shows that this rate is sharp up to poly-logarithmic factors.

Those results can readily be generalized to $d$-dimensionnal covariates. Assume that $m : [0, 1]^d \to [0, 1]$ is such that the weak Lipschitz assumption 2 holds for the euclidean distance, and that the margin assumption 3 is verified. Then, if $a_i \stackrel{i.i.d}{\sim} \mathcal{U}\left([0, 1]^d\right)$, we can adapt UCBF by dividing the space $[0, 1]^d$ into $K^d$ boxes of equal size. Looking more closely at our methods of proof, we note that the discretization error $R_T^{(d)}$ remains of order $O(T/K^2)$, while the cost of learning $R_T^{(FMAB)}$ is now bounded by $K^d \log(T) \log(K)$. Thus, the optimal number of intervals $K$ is of the order $T^{1/(d+2)} \log(T)^{-2/(d+2)}$, and the regret is of the order $O(T^{d/(d+2)} \log(T)^{4/(d+2)})$. We refer the interested reader to the Appendix, where precise statements of our hypotheses and results are provided, along with a description of the extension of Algorithm UCBF to multi-dimensional covariates.

## Broader impact

We present an algorithm for the problem of allocating a limited budget among competing candidates. This algorithm is easy to implement, and enjoys strong theoretical guarantees on its performance making it attractive and reliable in relevant applications. Nevertheless, we emphasise that the considered framework is based on the premise that the decision-maker is purely utility-driven. We leave it to the decision-maker to take additional domain-specific considerations into account.

## Acknowledgement

We would like to thank Christophe Giraud, Vianney Perchet and Gilles Stoltz for their stimulating suggestions and discussions.

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
