[Supplementary Material]

# Appendix

Theorem 1 is proved Section A, and Theorem 2 is proved Section B. Section C is dedicated to stating and proving an upper bound on the regret of UCBF in higher dimension. Lemmas used in those Sections are proved in Section D. First, let us state the following Lemma, which controls the fluctuations of $m$ within an interval.

**Lemma 2.** *Let $a \in [0, 1]$ be such that $m(a) = M + \alpha L/K$ for some $\alpha > 0$. Moreover, let $k$ be such that $a \in I_k$. Then*

$$\max_{a' \in I_k} m(a') \leq M + (\alpha + (\alpha \vee 1)) \frac{L}{K},$$

*and*

$$\min_{a' \in I_k} m(a') \geq M + \left(\alpha - \frac{(\alpha \vee 2)}{2}\right) \frac{L}{K}.$$

*Similarly, let $a \in [0, 1]$ be such that $m(a) = M - \alpha \frac{L}{K}$, where $\alpha > 0$. Moreover, let $k$ be such that $a \in I_k$. Then*

$$\min_{a' \in I_k} m(a') \geq M - (\alpha + (\alpha \vee 1)) \frac{L}{K},$$

*and*

$$\max_{a' \in I_k} m(a') \leq M - \left(\alpha - \frac{(\alpha \vee 2)}{2}\right) \frac{L}{K}.$$

# A    Proof of Theorem 1

To prove Theorem 1, we show that the regret $R_T$ can be decomposed as the sum of a discretization error term and of a term corresponding to the regret of pulling a game of finite bandit with $K$ arms. To do so, we introduce further notations.

Recall that for $k = 1, ..., K$, $m_k = K \int_{a \in I_k} m(a) da$ is the mean payment for pulling an arm uniformly in interval $I_k$. In order to avoid cumbersome notations for reordering the intervals, we assume henceforth (without loss of generality) that $\{m_k\}_{1 \leq k \leq K}$ is a decreasing sequence.

If we knew the sequence $\{m_k\}_{1 \leq k \leq K}$ but not the reward of the arms $m(a_i)$, a reasonable strategy would be to pull all arms in the fraction $p$ of the best intervals, and no arm in the remaining intervals. If all intervals contained the same number of arms $N/K$, we would pull all arms in the interval $I_1$, $I_2$, up to $I_f$, where $f = \lfloor pK \rfloor$, and we would pull the remaining arms randomly in $I_{f+1}$. Note however that since the arms are randomly distributed, the number of arms in each interval varies. Thus, a good strategy if we knew the sequence $\{m_k\}_{1 \leq k \leq K}$ would consist in pulling all arms in the intervals $I_1$, $I_2$, up to $I_{\widehat{f}}$, where $\widehat{f}$ is such that $N_1 + .. + N_{\widehat{f}} < T \leq N_1 + .. + N_{\widehat{f}+1}$, and pull the remaining arms in $I_{\widehat{f}+1}$. We call this strategy "oracle strategy for the discrete problem", and we denote it $\phi^d$. Recall that we denote by $\phi^*(t)$ the arm pulled at time $t$ by the oracle strategy, and by $\phi(t)$ the arm pulled at time $t$ by UCBF.

We decompose $R_T$ as follows :

$$
\begin{aligned}
R_T &= \sum_{t=1..T} m(a_{\phi^*(t)}) - \sum_{t=1..T} m(a_{\phi(t)}) \\
&= \sum_{t=1..T} m(a_{\phi^*(t)}) - \sum_{t=1..T} m(a_{\phi^d(t)}) + \sum_{t=1..T} m(a_{\phi^d(t)}) - \sum_{t=1..T} m(a_{\phi(t)}).
\end{aligned}
$$

Let $R_T^{(d)} = \sum_{t=1..T} m(a_{\phi^*(t)}) - \sum_{t=1..T} m(a_{\phi^d(t)})$. By definition, $R_T^{(d)}$ is the regret of the oracle stratgey for the discrete problem, and corresponds to a discretization error. We bound this term in Section A.1.

Let $R_T^{(FMAB)} = \sum_{t=1..T} m(a_{\phi^d(t)}) - \sum_{t=1..T} m(a_{\phi(t)})$ be the regret of our strategy against the oracle strategy for the discrete problem. $R_T^{(FMAB)}$ corresponds to the regret of the corresponding finite $K$-armed bandit problem. A bound on this term is obtained in Section A.2.

## A.1 Bound on the discretization error $R_T^{(d)}$ and proof of Lemma 1

To bound the discretization error $R_T^{(d)}$, we begin by controlling the deviation of $\widehat{f}$ and $m_{\widehat{f}}$ from their theoretical counterparts $f$ and $M$.

**Lemma 3.** *With probability at least $1 - 4e^{-\frac{2N}{K^2}}$, we have $|\widehat{f} - f| \leq 1$. On this event, $\left|m_{\widehat{f}} - M\right| \leq 4L/K$ and $m_{\widehat{f}+1} \in [M - 8L/K, M + L/K]$.*

Then, we define $\widehat{M} = m(a_{\phi*(T)})$ and control its deviation from $M$.

**Lemma 4.** *Assume that $p \in (1/K, 1 - 1/K)$. Then, with probability at least $1 - 2e^{-\frac{2N}{K^2}}$, we have $|\widehat{M} - M| \leq L/K$.*

We show later that with high probability, $\phi^*$ and $\phi^d$ may only differ on arms $i$ such that $m(a_i) \in [M - 16L/K, M + L/K]$. The following lemma controls the number of those arms.

**Lemma 5.** *Assume that $K \leq N^{2/3}$. Then, with probability at least $1 - 2e^{-\frac{N^{1/3}}{3}}$,*

$$\left|\left\{i : m(a_i) \in \left[M - \frac{16L}{K}, M + \frac{L}{K}\right]\right\}\right| \leq \frac{32LQN}{K}.$$

Using Lemmas 3-5, we control the discretization cost $R_T^{(d)}$ on the following event. Let

$$\begin{aligned}
\mathcal{E}_a &= \left\{|\widehat{f} - f| \leq 1\right\} \cap \left\{|\widehat{M} - M| \leq L/K\right\} \\
&\cap \left\{\left|\left\{i : m(a_i) \in \left[M - \frac{16L}{K}, M + \frac{L}{K}\right]\right\}\right| \leq \frac{32LQN}{K}\right\}.
\end{aligned}$$

Note that under the assumptions of Lemmas 4-5, $\mathbb{P}(\mathcal{E}_a) \geq 1 - 6e^{-\frac{2N}{K^2}} - 2e^{-\frac{N^{1/3}}{3}}$ by Lemma 3-5. Moreover on $\mathcal{E}_a$, $|m_{\widehat{f}} - M| \leq 4L/K$ and $m_{\widehat{f}+1} \in [M - 8L/K, M + L/K]$.

**Lemma 6.** *On the event $\mathcal{E}_a$, $R_T^{(d)} \leq \frac{384QL^2N}{K^2}$.*

Lemma 1 follows from Lemma 3, Lemma 4, Lemma 5 and Lemma 6.

## A.2 Bound on the regret of the discrete problem $R_T^{(FMAB)}$

We bound $R_T^{(FMAB)}$ on a favourable event, on which both the number of arms in each interval and the payment obtained by pulling those arms do not deviate too much from their expected value. Under Assumption 1, $\mathbb{E}[N_k] = N/K$ for all $k = 1, ..., K$. The following Lemmas provides a high probability bound on $\max_{k=1,...,K} |N_k - N/K|$.

**Lemma 7.** *Assume that $K \leq N^{2/3}/4$. Then,*

$$\mathbb{P}\left(\max_{k \in \{1,..,K\}} \left|N_k - \frac{N}{K}\right| \geq \frac{N}{2K}\right) \leq 2Ke^{-\frac{N^{1/3}}{3}}.$$

Now, we show that on an event of large probability, for $k = 1, ..., K$ and $s \leq (N_k \wedge T)$, $\widehat{m}_k(s)$ does not deviate of $m_k$ by more that $\sqrt{\log(T/\delta)/2s}$.

Let $k \in \{1, ..., K\}$ be such that $N_k > 0$. For $s \leq n_k(T)$, we denote by $\pi_k(s)$ the $s$-th armed pulled in interval $I_k$ by UCBF. With these notations, for all $s = 1, ..., n_k(T)$, $\widehat{m}_k(s)$ is defined by UCBF as $\widehat{m}_k(s) = \frac{1}{s} \sum_{t=i=1,...,s} y_{\pi_k(i)}$. We define similarly $\pi_k(s)$ for $s \in [n_k(T)+1, N_k]$ by selecting uniformly at random without replacement the remaining arms in $I_k$, and let $\widehat{m}_k(s) = \frac{1}{s} \sum_{t=i=1,...,s} y_{\pi_k(i)}$ for $s = n_k(T) + 1, ..., N_k$, and $\widehat{m}_k(0) = 0$.

**Lemma 8.**

$$\mathbb{P}\left(\exists k \in \{1, ..., K\}, s \leq (N_k \wedge T) : |\widehat{m}_k(s) - m_k| \geq \sqrt{\frac{\log(T/\delta)}{2s}}\right) \leq 2K\delta.$$

Then, we define

$$\mathcal{E}_b = \left\{\underset{k=1..K}{\cap}\left\{N_k \in \left[\frac{N}{2K},\frac{3N}{2K}\right]\right\}\right\}$$
$$\cap\left\{\underset{k=1..K}{\cap}\ \underset{s=1..(N_k \wedge T)}{\cap}\left\{|m_k - \widehat{m}_k(s)| \leq \sqrt{\frac{\log(T/\delta)}{2s}}\right\}\right\}.$$

Combining Lemma 7 and Lemma 8, we find that when $K \leq N^{2/3}/4$,

$$\mathbb{P}\left(\mathcal{E}_b\right) \geq 1 - 2Ke^{-\frac{N^{1/3}}{2}} - 2K\delta.$$

Now, we decompose $R_T^{(FMAB)}$ in the following way. Recall that

$$R_T^{(FMAB)} = \sum_{t=1...T} m(a_{\phi^d(t)}) - \sum_{t=1...T} m(a_{\phi(t)}).$$

Recall that $\phi^d$ pulls all arms in the interval $I_1, ..., I_f$, and pull the remaining arms in the interval $I_{f+1}$. In the following, we denote $\Phi^d(T)$ the set of arm pulled by $\phi^d$ at time $T$. Thus,

$$\sum_{t=1...T} m(a_{\phi^d(t)}) = \sum_{k=1..\widehat{f}}\ \sum_{a_i \in I_k} m(a_i) + \sum_{a_i \in \Phi^d(T) \cap I_{\widehat{f}+1}} m(a_i).$$

The number of arms pulled by $\phi^d$ is equal to $T$, and we can write

$$\sum_{t=1...T} m(a_{\phi^d(t)}) = \sum_{k=1..\widehat{f}}\ \sum_{a_i \in I_k} (m(a_i) - M) + \sum_{a_i \in \Phi^d(T) \cap I_{\widehat{f}+1}} (m(a_i) - M) + TM. \quad (5)$$

On the other hand, we decompose the total payment obtained by $\phi$ as the sum of the payment obtained by pulling arms also selected by $\phi^d$ (i.e. arms in $I_1, ..., I_f$ and $I_{f+1} \cap \Phi^d(T)$)), and the sum of payment for pulling arms that were not selected by $\phi^d$ (i.e. arms in $I_{f+1} \cap \overline{\Phi^d(T)}$ and in $I_{f+2}, ..., I_k$). Recall that $\Phi(T)$ is the set of arms pulled by UCBF at time $T$.

$$\sum_{t=1...T} m(a_{\phi(t)}) = \sum_{k=1..\widehat{f}}\ \sum_{a_i \in I_k \cap \Phi(T)} m(a_i) + \sum_{a_i \in I_{\widehat{f}+1} \cap \Phi(T) \cap \Phi^d(T)} m(a_i)$$
$$- \left(\sum_{a_i \in I_{\widehat{f}+1} \cap \Phi(T) \cap \overline{\Phi^d(T)}} - m(a_{\phi(t)})\right) - \left(\sum_{k=\widehat{f}+2..K}\ \sum_{a_i \in I_k \cap \Phi(T)} - m(a_{\phi^d(t)})\right).$$

Again, $T$ arms are pulled by $\phi$, and we can write

$$\sum_{t=1...T} m(a_{\phi(t)}) = \sum_{k=1..\widehat{f}}\ \sum_{a_i \in I_k \cap \Phi(T)} (m(a_i) - M) + \sum_{a_i \in I_{\widehat{f}+1} \cap \Phi(T) \cap \Phi^d(T)} (m(a_i) - M)$$
$$- \sum_{a_i \in I_{\widehat{f}+1} \cap \Phi(T) \cap \overline{\Phi^d(T)}} (M - m(a_i)) - \sum_{k=\widehat{f}+2..K}\ \sum_{a_i \in I_k \cap \Phi(T)} (M - m(a_i)) + TM.$$

$$(6)$$

Subtracting equation (6) from equation (5), we find that

$$R_T^{(FMAB)} = \sum_{k=1..\widehat{f}}\ \sum_{a_i \in I_k \cap \overline{\Phi(T)}} (m(a_i) - M) + \sum_{a_i \in I_{\widehat{f}+1} \cap \Phi^d(T) \cap \overline{\Phi(T)}} (m(a_i) - M)$$
$$+ \sum_{a_i \in I_{\widehat{f}+1} \cap \Phi(T) \cap \overline{\Phi^d(T)}} (M - m(a_i)) + \sum_{k=\widehat{f}+2..K}\ \sum_{a_i \in I_k \cap \Phi(T)} (M - m(a_i)).$$

We write

$$R_{\hat{f}+1} = \sum_{a_i \in I_{\hat{f}+1} \cap \Phi^d(T) \cap \overline{\Phi(T)}} (m(a_i) - M) + \sum_{a_i \in I_{\hat{f}+1} \cap \Phi(T) \cap \overline{\Phi^d(T)}} (M - m(a_i)),$$

$$R_{opt} = \sum_{k=1..\hat{f}} \sum_{a_i \in I_k \cap \overline{\Phi(T)}} (m(a_i) - M),$$

and

$$R_{subopt} = \sum_{k=\hat{f}+2..K} \sum_{a_i \in I_k \cap \Phi(T)} (M - m(a_i)).$$

The decomposition $R_T^{(FMAB)} = R_{opt} + R_{\hat{f}+1} + R_{subopt}$ show that three phenomenons contribute to the regret of $\phi$ on the discrete problem. The side effect term $R_{\hat{f}+1}$ can easily be bounded : there are most $1.5N/K$ arms in $I_{\hat{f}+1}$, and so there are at most $1.5N/K$ terms in $R_{\hat{f}+1}$. On the event $\mathcal{E}_a$, $m_{\hat{f}+1} \in [M - 8L/K, M + L/K]$. Using Lemma 2, we see that for each arm $a_i \in I_{\hat{f}+1}$, $|m(a_i) - M| \leq 16L/K$. Thus, on $\mathcal{E}_a \cap \mathcal{E}_b$, $R_{\hat{f}+1} \leq 24N/K^2$.

Now, we say that an interval $I_k$ is sub-optimal if $m_k < m_{\hat{f}+1}$ and is optimal if $m_k \geq m_{\hat{f}}$. $R_T^{(FMAB)} - R_{\hat{f}+1}$ is the sum of a term $R_{opt}$, induced by the remaining arms in the optimal intervals, and a term $R_{subopt}$, induced by pulls of arms in sub-optimal intervals. The following Lemma will be used to control those terms.

For two intervals $I_k$, $I_l$ such that $m_k > m_l$, we provide a bound on the number of arms drawn in $I_l$ given that there are still arms available in the better interval $I_k$. For two intervals $k, l \in \{1, ..., K\}^2$, we denote henceforth $\Delta_{k,l} = m_k - m_l$.

**Lemma 9.** *Let $k \in \{1, ..., K\}$. On the event $\mathcal{E}_b \cap \{n_k(T) < N_k\}$, a.s. for all intervals $I_l$ such that $\Delta_{k,l} > 0$, $n_l(T) \leq \frac{3\log(T/\delta)}{\Delta_{k,l}^2}$.*

To bound the regret $R_{subopt}$, we take advantage of the fact that every slightly sub-optimal interval $k$ cannot be selected more than $N_k$ times. This is done in the following lemma.

**Lemma 10.** *On the event $\mathcal{E}_a \cap \mathcal{E}_b$,*

$$R_{subopt} \leq \frac{600L^2QN}{K^2} + 384\log(T/\delta)KQ\left(\log_2(K/L) \vee 1\right).$$

While $R_{subopt}$ corresponds to the regret of pulling sub-optimal arms, and is bounded using classical bandit arguments, $R_{opt}$ corresponds to the regret of not having pulled optimal arms. We first control the number of optimal arms that have not been pulled. The arguments used to prove Lemma 10 can be used to control the number of arms pulled in sub-optimal intervals, which is equal to the number of non-zero terms in $R_{opt}$.

**Lemma 11.** *On $\mathcal{E}_a \cap \mathcal{E}_b$, the number of arms pulled in sub-optimal intervals by UCBF is bounded by $30Q(LN/K + \log(T/\delta)K^2/L)$.*

This number is equal to the number of optimal arms that have not been pulled, and thus to the number of non-zero terms in $R_{opt}$. Note that this number is at least of order $N/K \vee K^2$, while $R_T^{(d)} + R_{subopt}$ is of the order $N/K^2 \vee K$ . Thus, bounding each term in $R_{opt}$ by 1 will likely lead to sub-optimal bounds on the regret $R_T$. In the next Lemma, we characterise intervals whose arms have all been pulled by UCBF. Note that those intervals do not contributes to $R_{opt}$.

**Lemma 12.** *Let $A = 35\sqrt{\frac{K^3Q\log(T/\delta)}{NL} \vee 1}$. At time $T$, on the event $\mathcal{E}_a \cap \mathcal{E}_b$, all arms in intervals $I_k$ such that $m_k \geq M + AL/K$ have been pulled.*

Using Lemmas 11 and 12, we can finally control $R_{opt}$.

**Lemma 13.** *On event $\mathcal{E}_a \cap \mathcal{E}_b$,*

$$R_{opt} \leq 60AQ\left(\frac{L^2N}{K^2} + \log(T/\delta)K\right)$$

To conclude, note that for the choice $\delta = N^{-4/3}$ and $K = \lfloor N^{1/3} \log(N)^{-2/3} \rfloor \geq (1 - p \wedge p)^{-1}$,

$$A \leq 35\sqrt{\frac{N \log(N)^{-2} \log(pN^{7/3})}{NL} \vee 1} \leq 35\sqrt{\frac{7 \log(N)^{-1}}{3L} \vee 1}.$$

Moreover $K^{-2} = \lfloor N^{1/3} \log(N)^{-2/3} \rfloor^{-2} \leq 4 \left( N^{1/3} \log(N)^{-2/3} \right)^{-2}$ since $N^{1/3} \log(N)^{-2/3} \geq 2$. Thus, on the event $\mathcal{E}_a \cap \mathcal{E}_b$,

$$
\begin{aligned}
R_{opt} &\leq 2100Q \left( 4L^2 N^{1/3} \log(N)^{4/3} + 7/3 \log(N)^{1/3} N^{1/3} \right) \sqrt{\frac{7 \log(N)^{-1}}{3L} \vee 1} \\
R_{subopt} &\leq 2400L^2 Q N^{1/3} \log(N)^{4/3} + 896 Q N^{1/3} \log(N)^{1/3} \left( \log_2(N/L) \vee 1 \right) \\
R_{\hat{f}+1} &\leq 96 N^{1/3} \log(N)^{4/3} \\
R_T^{(d)} &\leq 1536 Q L N^{1/3} \log(N)^{4/3}.
\end{aligned}
$$

Thus, on $\mathcal{E}_a \cap \mathcal{E}_b$, we find that

$$R_T \leq C N^{1/3} \log(N)^{4/3},$$

or equivalently that

$$R_T \leq C (T/p)^{1/3} \log(T/p)^{4/3}$$

for some constant $C$ depending only on $L$ and $Q$. Note that $K \leq N^{2/3}/4$ as soon as $N \geq 30$. Using the Lemmas 3, 4, 5, 7 and 8, we find that the event $\mathcal{E}_a \cap \mathcal{E}_b$ occurs with probability at least $1 - 6^{-2\lfloor N^{1/3} \log(N)^{4/3} \rfloor} - 2e^{-N^{1/3}/3} - 2e^{-N^{1/3}/3} N^{1/3} \log(N)^{-2/3} - 2N^{-1} \geq 1 - 12(N^{-1} \vee e^{-N^{1/3}/3})$.

# B  Proof of Theorem 2

Before proving Theorem 2, we recall that under Assumption 5, the set of covariates $(a_1, ..., a_N) = (1/N, ..., 1)$ is deterministic. We prove Theorem 2 by studying reward that are independent Bernoulli variables : under Assumption 4, $y_i \sim \text{Bernoulli}(m(a_i))$ for $i = 1, .., N$. At each time $t$, a strategy $\phi$ selects which arm $\phi(t)$ to pull based on the past observations $(\phi(1), y_{\phi(1)}, ..., \phi(t-1), y_{\phi(t-1)})$. For $t = 1, ..., T$, let $\mathcal{H}_t = (a_1, y_1, ..., a_t, y_t)$.

Let $m_0$ and $m_1$ be two payoff functions. We denote by $\mathbb{P}_0$ the distribution of $\mathcal{H}_T$ when the payoff function is $m_0$, and $\mathbb{P}_1$ the distribution of $\mathcal{H}_T$ when the payoff function is $m_1$. Moreover, let $\bar{\mathcal{Z}}$ be any event $\sigma(\mathcal{H}_T)$-measurable. According to Bretagnolle-Huber inequality (see, e.g., Theorem 14.2 in [Lattimore and Szepesvári, 2020])

$$\mathbb{P}_0(\mathcal{Z}) + \mathbb{P}_1(\bar{\mathcal{Z}}) \geq \frac{1}{2} \exp\left( -KL(\mathbb{P}_0, \mathbb{P}_1) \right).$$

Let us sketch the proof of Theorem 2. In a first time, we design two payoff functions $m_0$ and $m_1$ that satisfy Assumptions 2 and 3 and differ on a small number of arms. Then, we bound their Kullback-Leibler divergence. Finally, we define an event $\mathcal{Z}$ which is favorable for $m_1$ and unfavorable for $m_0$, and we provide lower bounds for $R_T$ on $\mathcal{Z}$ under $\mathbb{P}_0$ and on $\bar{\mathcal{Z}}$ under $\mathbb{P}_1$.

We will henceforth assume that

$$N \geq \frac{1}{(p \wedge 1 - p)^3 (L \wedge 0.5)^2} \vee 811.$$

In order to define $m_0$ and $m_1$, we introduce the following notations. Let $\alpha \in (20N^{-2/3}, 0.5]$ to be defined later, and let $\tilde{L} = L \wedge 0.5$ and $\delta = \alpha(N\tilde{L}^2)^{-1/3}$. Now, define $x_0 = 1 - p - 2\delta$ and $x_1 = 1 - p + 2\delta$. The inequality $2\delta < p \wedge (1 - p)$ ensures that $0 < x_0 < 1 - p < x_1 < 1$. Moreover, $\tilde{L}(x_0 \vee 1 - x_1) \leq 1/2$ and $\tilde{L}\delta < 1/4$. We define $m_0$ and $m_1$ as follows.

$$
m_0(x) = \begin{cases}
1/2 - \tilde{L}(x_0 - x) & \text{if } x \in [0, x_0) \\
1/2 - \tilde{L}(x - x_0) & \text{if } x \in [x_0, x_0 + \delta) \\
1/2 - \tilde{L}(1 - p - x) & \text{if } x \in [x_0 + \delta, 1 - p) \\
1/2 + \tilde{L}(x - (1 - p)) & \text{if } x \in [1 - p, 1 - p + \delta) \\
1/2 + \tilde{L}(x_1 - x) & \text{if } x \in [1 - p + \delta, x_1) \\
1/2 + \tilde{L}(x - x_1) & \text{if } x \in [x_1, 1]
\end{cases}
$$

Define similarly

$$
m_1(x) = \begin{cases}
1/2 - \tilde{L}(x_0 - x) & \text{if } x \in [0, x_0) \\
1/2 + \tilde{L}(x - x_0) & \text{if } x \in [x_0, x_0 + \delta) \\
1/2 + \tilde{L}(1 - p - x) & \text{if } x \in [x_0 + \delta, 1 - p) \\
1/2 - \tilde{L}(x - (1 - p)) & \text{if } x \in [1 - p, 1 - p + \delta) \\
1/2 - \tilde{L}(x_1 - x) & \text{if } x \in [1 - p + \delta, x_1) \\
1/2 + \tilde{L}(x - x_1) & \text{if } x \in [x_1, 1]
\end{cases}
$$

The functions $m_0$ and $m_1$ are bounded in $[0, 1]$, piecewise linear. They differ only on $[x_0, x_1]$, and are such that

$$
\min \{A : \lambda (\{x : m_0(x) \geq A\}) < p\} = \min \{A : \lambda (\{x : m_1(x) \geq A\}) < p\} = 1/2.
$$

Under hypotesis 5, the $T = pN$ best arms for the payoff function $m_0$ are in $[1 - p, 1] \cap \{x_0\}$, while the $T = pN$ best arms for the payoff function $m_1$ are in $[x_1, 1] \cap [x_0, 1 - p]$.

**Lemma 14.** *The payoff functions $m_0$ and $m_1$ satisfy Assumptions 2 and 3.*

Next, we bound the Kullback-Leibler divergence between $\mathbb{P}_0$ and $\mathbb{P}_1$.

**Lemma 15.** *For the functions $m_0$ and $m_1$ defined above,*

$$
KL(\mathbb{P}_0, \mathbb{P}_1) \leq 70.4\alpha^3.
$$

We define $\mathcal{Z}$ as the following event :

$$
\mathcal{Z} = \left\{ \sum_{a_i \in [x_0, 1 - p]} \mathbb{1}_{\{i \in \Phi(T)\}} \geq N\delta - 2 \right\}.
$$

Because of Assumption 5, there are between $\lfloor 2N\delta \rfloor$ and $\lceil 2N\delta \rceil$ arms in $(x_0, 1 - p)$. Under $\mathbb{P}_0$, the arms in $(x_0, 1 - p)$ are sub-optimal, so $\mathcal{Z}$ is disadvantageous. On the contrary, under $\mathbb{P}_1$ all arms in $(x_0, 1 - p)$ are optimal under $m_1$, and so $\overline{\mathcal{Z}}$ is disadvantageous. We provide a more detailed statement in the following lemma.

**Lemma 16.** *Under $\mathbb{P}_0$, on $\mathcal{Z}$, $R_T \geq 0.22\alpha^2 N^{1/3}$. Under $\mathbb{P}_1$, on $\overline{\mathcal{Z}}$, $R_T \geq 0.22\alpha^2 N^{1/3}$.*

Since $N \geq 811$, we can choose for example $\alpha = 0.23$. Using $(a \vee b) \geq (a + b)/2$, we see that

$$
\max \left\{ \mathbb{P}_0 \left( R_T \geq 0.01 N^{1/3} \right), \mathbb{P}_1 \left( R_T \geq 0.01 N^{1/3} \right) \right\} \geq 0.1.
$$

## C   Upper bound on the regret in multi-dimensional settings

In this section, we provide an upper bound on the regret of a natural extension of Algorithm UCBF to $d$-dimensional covariates. More precisely, we assume that the arms are described by covariates in the set $\mathcal{X} = [0, 1]^d$ for some $d \in \mathbb{N}^*$. Similarly to the one-dimensional case, we assume that the covariates are uniformly distributed in $\mathcal{X}$:

**Assumption 6.** *For $i = 1, ..., N$, $a_i \overset{i.i.d.}{\sim} \mathcal{U}([0, 1]^d)$.*

As in the one dimensional setting, we assume that the mean payoff function is weakly $L$-Lipschitz with regard to the Euclidean distance:

**Assumption 7.** *For all* $(x, y) \in [0, 1]^d \times [0, 1]^d$,

$$|m(x) - m(y)| \leq \max\left\{|M - m(x)|, L\|x - y\|_2\right\}.$$

Moreover we assume that the mean reward function $m : [0, 1]^d \to [0, 1]$ verifies Assumption 3 (here, $\lambda$ denotes the Lebesgue measure on $[0, 1]^d$). Then, the UCBF Algorithm can readily be generalized to this $d$-dimensional setting, as described in Algorithm 2. The following Theorem bounds the regret of Algorithm $d$-UCBF.

---

**Algorithm 2** $d$-dimensional Upper Confidence Bound for Finite continuum-armed bandits ($d$-UCBF)

---

**Parameters:** $K, \delta$

**Initialisation:** Divide $[0, 1]^d$ into $K^d$ bins $B_k$ such that for $k \in \{0, ..., K^d - 1\}$, $B_k = [\frac{k_1}{K}, \frac{k_1+1}{K}) \times ... \times [\frac{k_d}{K}, \frac{k_d+1}{K})$, where $(k_1, ..., k_d)$ denotes the $d$-ary representation of $k$. Let $N_k = \sum_{1 \leq i \leq N} \mathbb{1}\{a_i \in B_k\}$ be the number of arms in the bin $B_k$. Define the set of bins alive as the set of bins $B_k$ such that $N_k \geq 2$. Pull an arm uniformly at random in each bin alive.

**for** $t = K^d + 1, ..., T$ **do**

   − Select an bin $B_k$ that maximizes $\widehat{m}_k(n_k(t-1)) + \sqrt{\frac{\log(T/\delta)}{2n_k(t-1)}}$ among the set of alive bins, where $n_k(t-1)$ is the number of arms pulled from $B_k$ by the algorithm before time $t$, and $\widehat{m}_k(n_k(t-1))$ is the average reward obtained from those $n_k(t-1)$ samples.

   − Pull an arm selected uniformly at random among the arms in $B_k$. Remove this arm from $B_k$. If $B_k$ is empty, remove $B_k$ from the set of alive bins.

**end for**

---

**Theorem 3.** *Under Assumption 6, 7 and 3, there exists a constant $C_{L,Q,p,d}$ depending only on $L, Q$ $p$ and $d$ such that for the choice $K = \lceil N^{\frac{1}{d+2}} \log(N)^{-\frac{2}{d+2}} \rceil$ and $\delta = N^{-\frac{2d+2}{d+2}}$,*

$$R_T \leq C_{L,Q,p,d} \, T^{\frac{d}{d+2}} \log(T)^{\frac{4}{d+2}}$$

*with probability $1 - O(N^{-1})$.*

The rest of this Section is devoted to proving Theorem 3. To do so, we follow the main lines of the proof of Theorem 1. Some Lemmas follow readily from results developed in Section A, and their proofs are therefore omitted. The remaining Lemmas are proved in Section D.

Let us now prove Theorem 3. As for Theorem 1, we begin by controlling the fluctuations of the mean payoff function $m$ within a bin.

**Lemma 17.** *Let $a \in [0, 1]^d$ be such that $m(a) = M + \alpha L/K$ for some $\alpha > 0$. Moreover, let $k$ be such that $a \in B_k$. Then*

$$\max_{a' \in B_k} m(a') \leq M + \left(\alpha + (\alpha \vee \sqrt{d})\right)\frac{L}{K},$$

*and*

$$\min_{a' \in B_k} m(a') \geq M + \left(\alpha - \frac{(\alpha \vee 2\sqrt{d})}{2}\right)\frac{L}{K}.$$

*Similarly, let $a \in [0, 1]^d$ be such that $m(a) = M - \alpha\frac{L}{K}$, where $\alpha > 0$. Moreover, let $k$ be such that $a \in B_k$. Then*

$$\min_{a' \in B_k} m(a') \geq M - \left(\alpha + (\alpha \vee \sqrt{d})\right)\frac{L}{K},$$

*and*

$$\max_{a' \in B_k} m(a') \leq M - \left(\alpha - \frac{(\alpha \vee 2\sqrt{d})}{2}\right)\frac{L}{K}.$$

*Proof.* In the general $d$-dimensional case, two points in the same bin may be separated by a Euclidean distance of $\sqrt{d}/K$. Using this remark, one can readily adapt the proof of Lemma 2 to prove Lemma 17. □

Conversely, we obtain a lower bound on the Lebesgue measure of arms with mean reward close to $M$.

**Lemma 18.** *There exist a constant $c_{p,d} > 0$ depending only on $p$ and $d$ such that for all $t \in (0, \sqrt{d}L]$,*

$$\mathbb{P}\left(m(a_1) \in [M, M+t)\right) \geq c_{p,d}\frac{t}{L}.$$

Then, we decompose the regret $R_T$ into the sum of a discretization error, and of the cost of learning in the corresponding finite $K^d$-armed bandit problem. For $k = 0, ..., K^d - 1$, we define $m_k = K^d \int_{a \in B_K} m(a)da$ as the mean payment for pulling an arm uniformly in bin $B_k$. In order to avoid cumbersome notations for reordering the bins, we assume henceforth (without loss of generality) that $\{m_k\}_{0 \leq k \leq K^d-1}$ is a decreasing sequence. Similarly to the one-dimensional case, we denote by $\phi^d$ the strategy pulling all arms in the bin $B_1$, $B_2$, up to $B_{\widehat{f}}$ and pulling the remaining arms in $B_{\widehat{f}+1}$, where $\widehat{f}$ is such that $N_1 + .. + N_{\widehat{f}} < T \leq N_1 + .. + N_{\widehat{f}+1}$. Note that $\phi^d$ corresponds to the oracle strategy for the discretized problem. We also denote $f = \lfloor pK^d \rfloor$. Recall that we denote by $\phi^*(t)$ the arm pulled at time $t$ by the oracle strategy, and by $\phi(t)$ the arm pulled at time $t$ by UCBF.

Now, decompose $R_T$ as follows :

$$\begin{aligned}
R_T &= \sum_{t=1..T} m(a_{\phi^*(t)}) - \sum_{t=1..T} m(a_{\phi(t)}) \\
&= \sum_{t=1..T} m(a_{\phi^*(t)}) - \sum_{t=1..T} m(a_{\phi^d(t)}) + \sum_{t=1..T} m(a_{\phi^d(t)}) - \sum_{t=1..T} m(a_{\phi(t)}).
\end{aligned}$$

Again, we denote by $R_T^{(d)} = \sum_{t=1..T} m(a_{\phi^*(t)}) - \sum_{t=1..T} m(a_{\phi^d(t)})$ the discretization error. Moreover, we define $R_T^{(FMAB)} = \sum_{t=1..T} m(a_{\phi^d(t)}) - \sum_{t=1..T} m(a_{\phi(t)})$ the regret of our strategy against the oracle strategy for the discrete problem.

As in the one-dimensional case, we use the following Lemmas to bound the discretization error $R_T^{(d)}$.

**Lemma 19.** *Define $\epsilon = \lceil c_{p,d}K^{d-1} \rceil$ and $\alpha = 4QL/c_{p,d} + 2/\sqrt{d} \times (1 + 3/K^{d-1})$, where $c_{p,d}$ is the constant appearing in Lemma 18. With probability at least $1 - 4\exp\left(-\frac{2c_{p,d}^2 N}{K^2}\right)$, we have $|\widehat{f} - f| \leq 1 + \epsilon$. On this event, $\left|m_{\widehat{f}} - M\right| \leq \alpha\sqrt{d}L/K$ and $\left|m_{\widehat{f}+1} - M\right| \leq \alpha\sqrt{d}L/K$.*

**Lemma 20.** *For the constant $c_{p,d} > 0$ defined in Lemma 18,*

$$\mathbb{P}\left(|\widehat{M} - M| \leq \sqrt{d}L/K\right) \geq 1 - 2e^{-\frac{2c_{p,d}^2 N}{K^2}}. \tag{7}$$

The proof of Lemma 20 is obtained by following the lines of the proof of Lemma 4, and applying Lemma 18. It is therefore omitted.

**Lemma 21.** *With probability at least $1 - 2\exp\left(-\frac{8\alpha^2 dL^2 Q^2 N}{K^2}\right)$,*

$$\left|\left\{i : m(a_i) \in \left[M - \frac{2\alpha\sqrt{d}L}{K}, M + \frac{L\sqrt{d}}{K}\right]\right\}\right| \leq \frac{4\alpha\sqrt{d}LQN}{K}.$$

The proof of Lemma 21 follows from the arguments developed in the proof of Lemma 5, and is therefore omitted. Note that since $LQ \geq 1$, $d \geq 1$ and $\alpha \geq 1 \geq c_{p,d}$, $1 - 2\exp\left(-\frac{8\alpha^2 dL^2 Q^2 N}{K^2}\right) \geq 1 - 2\exp\left(-\frac{2c_{p,d}^2 N}{K^2}\right)$.

**Lemma 22.** *Let*

$$\begin{aligned}
\mathcal{E}_a &= \left\{|\widehat{f} - f| \leq 1 + \epsilon\right\} \cap \left\{|\widehat{M} - M| \leq \sqrt{d}L/K\right\} \\
&\cap \left\{\left|\left\{i : m(a_i) \in \left[M - \frac{2\alpha\sqrt{d}L}{K}, M + \frac{\sqrt{d}L}{K}\right]\right\}\right| \leq \frac{4\alpha\sqrt{d}LQN}{K}\right\}.
\end{aligned}$$

*On the event $\mathcal{E}_a$, $R_T^{(d)} \leq \frac{8\alpha^2 dQL^2 N}{K^2}$.*

Combing Lemmas 19, 20 and 21, we note that $\mathbb{P}(\mathcal{E}_a) \geq 1 - 8\exp(\frac{2c_{p,d}^2 N}{K^2})$. Next, we bound the cost of learning on the corresponding finite $K^d$-armed bandit problem. Similarly to the one-dimensional case, we use the following Lemmas to control this term.

**Lemma 23.**
$$\mathbb{P}\left(\max_{k \in \{0,..,K^d-1\}} \left|N_k - \frac{N}{K^d}\right| \geq \frac{N}{2K^d}\right) \leq 2K^d e^{-\frac{N}{10K^d}}.$$

**Lemma 24.**
$$\mathbb{P}\left(\exists k \in \{0,...,K^d-1\}, s \leq (N_k \wedge T) : |\widehat{m}_k(s) - m_k| \geq \sqrt{\frac{\log(T/\delta)}{2s}}\right) \leq 2K^d\delta.$$

The proof of Lemma 24 follows closely the proof of Lemma 8, and is therefore omitted.

Now, we define
$$\mathcal{E}_b = \left\{\bigcap_{k=0,..,K^d-1} \left\{N_k \in \left[\frac{N}{2K^d}, \frac{3N}{2K^d}\right]\right\}\right\}$$
$$\cap \left\{\bigcap_{k=0,..,K^d-1} \bigcap_{s=1..(N_k \wedge T)} \left\{|m_k - \widehat{m}_k(s)| \leq \sqrt{\frac{\log(T/\delta)}{2s}}\right\}\right\}.$$

Combining Lemma 23 and Lemma 8, we find that
$$\mathbb{P}(\mathcal{E}_b) \geq 1 - 2K^d e^{-\frac{N}{10K^d}} - 2K^d\delta.$$

For two bins $k, l \in \{1, ..., K\}^2$, we denote henceforth $\Delta_{k,l} = m_k - m_l$.

**Lemma 25.** *Let $k \in \{1, ..., K\}$. On the event $\mathcal{E}_b \cap \{n_k(T) < N_k\}$, a.s. for all bins $B_l$ such that $\Delta_{k,l} > 0$, $n_l(T) \leq \frac{3\log(T/\delta)}{\Delta_{k,l}^2}$.*

The proof of Lemma 25 can be obtained by following the lines of the proof of Lemma 7, and is therefore omitted.

As in the one-dimensional case, we write $R_T^{(FMAB)} = R_{opt} + R_{\hat{f}+1} + R_{subopt}$, where
$$R_{\hat{f}+1} = \sum_{a_i \in B_{\hat{f}+1} \cap \Phi^d(T) \cap \overline{\Phi(T)}} (m(a_i) - M) + \sum_{a_i \in B_{\hat{f}+1} \cap \Phi(T) \cap \overline{\Phi^d(T)}} (M - m(a_i)),$$
$$R_{opt} = \sum_{k=1..\hat{f}} \sum_{a_i \in B_k \cap \overline{\Phi(T)}} (m(a_i) - M),$$
and
$$R_{subopt} = \sum_{k=\hat{f}+2..K^d-1} \sum_{a_i \in B_k \cap \Phi(T)} (M - m(a_i)).$$

The term $R_{\hat{f}+1}$ can easily be bounded : there are most $1.5N/K^d$ arms in $B_{\hat{f}+1}$, and so there are at most $1.5N/K^d$ terms in $R_{\hat{f}+1}$. On the event $\mathcal{E}_a$, $m_{\hat{f}+1} \in [M - \alpha\sqrt{d}L/K, M + \alpha\sqrt{d}L/K]$. Using Lemma 2, we see that for each arm $a_i \in B_{\hat{f}+1}$, $|m(a_i) - M| \leq 2\alpha\sqrt{d}L/K$. Thus, on $\mathcal{E}_a \cap \mathcal{E}_b$, $R_{\hat{f}+1} \leq 3\alpha\sqrt{d}LN/K^{d+1}$.

The following Lemmas help us bound the terms $R_{subopt}$ and $R_{opt}$.

**Lemma 26.** *On the event $\mathcal{E}_a \cap \mathcal{E}_b$,*
$$R_{subopt} \leq 120\alpha^2 dL^2 Q\left(\frac{N}{K^2} + \frac{K^d\log(T/\delta)\log_2(K/\alpha\sqrt{d}L)}{\alpha^2 dL^2}\right).$$

**Lemma 27.** *On $\mathcal{E}_a \cap \mathcal{E}_b$, the number of arms pulled in sub-optimal bins by UCBF is bounded by* $6\alpha\sqrt{d}LQN/K + 24\log(T/\delta)K^{d+1}Q/(\alpha\sqrt{d}L)$.

**Lemma 28.** *Let*

$$A = \sqrt{\frac{472QK^{d+2}\log(T/\delta)}{Nc_{p,d}Ld}} \vee 16\alpha QL/c_{p,d}.$$

*At time $T$, on the event $\mathcal{E}_a \cap \mathcal{E}_b$, all bins $B_k$ such that $m_k \geq M + A\sqrt{d}L/K$ have died.*

Combining Lemmas 26, 27 and 28, we prove the following result.

**Lemma 29.** *On event $\mathcal{E}_a \cap \mathcal{E}_b$,*

$$R_{opt} \leq 30\alpha AdL^2 Q \left( \frac{N}{K^2} + \frac{\log(T/\delta)K^d}{\alpha^2 dL^2} \right).$$

The proof of Lemma 29 is similar to that of Lemma 13, and is therefore omitted.

Thus, on the event $\mathcal{E}_a \cap \mathcal{E}_b$,

$$R_T \leq \left( 33\alpha AdL^2 Q + 128\alpha^2 dL^2 Q \right) \left( \frac{N}{K^2} + \frac{\log(T/\delta)\log_2(K/\alpha\sqrt{d}L)K^d}{\alpha^2 dL^2} \right).$$

The event $\mathcal{E}_a \cap \mathcal{E}_b$ happens with probability larger than $1 - 8\exp(\frac{2c_{p,d}^2 N}{K^2}) - 2K^d \exp(-\frac{N}{10K^d}) - 2K^d\delta$.
For the choice $K = \lceil N^{\frac{1}{d+2}}\log(N)^{-\frac{2}{d+2}} \rceil$ and $\delta = N^{-\frac{2d+2}{d+2}}$,

$$\begin{aligned}
\mathbb{P}(\mathcal{E}_a \cap \mathcal{E}_b) &\geq 1 - 8\exp\left( -2c_{p,d}^2 N^{\frac{d}{d+2}}\log(N)^{\frac{4}{d+2}} \right) - 2(N^{\frac{1}{d+2}}\log(N)^{\frac{-2}{d+2}} + 1)^d \log(N)^{\frac{-2d}{d+2}} \exp\left( -N^{\frac{2}{d+2}}\log(N)^{\frac{2d}{d+2}}/10 \right) \\
&\quad + 2(N^{\frac{1}{d+2}}\log(N)^{\frac{-2}{d+2}} + 1)^d N^{\frac{-(2d+2)}{d+2}} \\
&\geq 1 - O(N^{-1}).
\end{aligned}$$

Note that for this choice of $K$, $A$ is bounded by a constant depending on $\alpha$, $Q$, $L$ and $c_{p,d}$. Then, $\mathcal{E}_a \cap \mathcal{E}_b$, there exists a constant $C$ depending on $d$, $L$, $Q$ and $p$ such that

$$R_T \leq C \left( N^{\frac{d}{d+2}}\log(N)^{\frac{4}{d+2}} + \frac{\frac{3+2d}{(d+2)^2}\log(N)\log(N)(N^{\frac{1}{d+2}}\log(N)^{\frac{-2}{d+2}} + 1)^d}{\alpha^2 dL^2} \right).$$

This concludes the proof of Theorem 3.

# D   Proofs of auxiliary Lemmas

## D.1   Proof of Lemma 2

Recall that $a \in I_k$ and $\alpha > 0$ is such that $m(a) = M + \alpha L/K$. By Assumption 2, we see that for any $a' \in I_k$,

$$|(M + \alpha L/K) - m(a')| \leq \max\{\alpha L/K, L/K\},$$

so

$$m(a') \leq M + (\alpha + (\alpha \vee 1))L/K.$$

This yield the first part of the Lemma. To obtain the second part, note that Assumption 2 also implies

$$|m(a') - (M + \alpha L/K)| \leq \max\{|m(a') - M|, L/K\}.$$

Thus,

$$m(a') \geq M + \alpha L/K - \max\{|m(a') - M|, L/K\}. \tag{8}$$

If $|m(a') - M| \geq L/K$, then equation (8) implies

$$m(a') \geq M + \alpha L/K - (m(a') - M).$$

Thus,

$$2m(a') \geq 2M + \alpha L/K$$

and

$$m(a') \geq M + \frac{\alpha}{2}L/K.$$

Since $|m(a') - M| = m(a') - M = \alpha L/(2K)$, $|m(a') - M| \geq L/K$ implies $\alpha \geq 2$. On the other hand, if $|m(a') - M| < L/K$, equation 8 implies

$$
\begin{aligned}
m(a') &\geq M + \alpha L/K - L/K \\
&\geq M + \frac{(\alpha - 1)L}{K}.
\end{aligned}
$$

Since $m(a') - M \leq |m(a') - M|$, the assumption $|m(a') - M| < L/K$ implies that $\alpha < 2$.

To summarise, when $\alpha < 2$ we necessarily have $|m(a') - M| < L/K$, and $m(a') \geq M + (\alpha - 1)L/K$. On the contrary, when $\alpha \geq 2$ we necessarily have $|m(a') - M| \geq L/K$, and $m(a') \geq M + \alpha L/(2K)$. This writes

$$m(a') \geq M + \left( \alpha - \frac{\alpha \vee 2}{2} \right) \frac{L}{K}.$$

Using the same arguments, we can prove similar bounds for the case $m(a) = M - \alpha L/K$.

## D.2 Proof of Lemma 3

Recall that $f = \lfloor pK \rfloor$, and $\hat{f}$ is such that $N_1 + .. + N_{\hat{f}} < T \leq N_1 + .. + N_{\hat{f}+1}$. By definition, $N_1 + .. + N_{f-1} = \sum_{1 \leq i \leq N} \mathbb{1}_{\{a_i \in I_1 \cup .. \cup I_{f-1}\}}$, where $\mathbb{1}_{\{a_i \in I_1 \cup .. \cup I_{f-1}\}}$ are independant Bernoulli random variables of parameter $\frac{f-1}{K}$. Using Hoeffding's inequality, we find that

$$\mathbb{P}\left( \sum_{1 \leq i \leq N} \mathbb{1}_{\{a_i \in I_1 \cup .. \cup I_{f-1}\}} - \frac{(f-1)N}{K} \geq \frac{N}{K} \right) \leq e^{-\frac{2N}{K^2}}.$$

Now, by definition, $f = \lfloor TK/N \rfloor$, and so $fN/K \leq T$. Thus,

$$\mathbb{P}\left( N_1 + .. + N_{f-1} \geq T \right) \leq e^{-\frac{2N}{K^2}}. \tag{9}$$

This shows that with high probability, $N_1 + .. + N_{f-1} < T$, which implies that $f - 1 < \hat{f} + 1$. Using again Hoeffding's inequality, we find that

$$\mathbb{P}\left( \frac{(f+2)N}{K} - \sum_{1 \leq i \leq N} \mathbb{1}_{\{a_i \in I_1 \cup .. \cup I_{f+2}\}} \geq \frac{N}{K} \right) \leq e^{-\frac{2N}{K^2}}.$$

By definition of $f$, $(f+1)N/K \geq T$. Thus,

$$\mathbb{P}\left( N_1 + .. + N_{f+2} \geq T \right) \leq e^{-\frac{2N}{K^2}} \tag{10}$$

This shows that with high probability, $T < N_1 + .. + N_{f+2}$, and thus $f + 2 > \hat{f}$. Combining equations (16) and (10), we find that with probability larger than $1 - 2e^{-\frac{2N}{K^2}}$, $|f - \hat{f}| \leq 1$.

In a second time, we prove that $m_f \in [M - L/K, M + L/K]$. To do so, we first show that there are at least $\lceil pK \rceil$ intervals $k$ such that $m_k \geq M - L/K$, or equivalently that there are at most $\lfloor (1-p)K \rfloor$ intervals $k$ such that $m_k < M - L/K$. Indeed, for all $k$ such that $m_k < M - L/K$, there exists $a \in I_k$ such that $m(a) < M - L/K$. Using Lemma 2, we see that $\forall a \in I_k, m(a) \leq M$.

By definition of $p$, there can be at most $\lfloor(1-p)K\rfloor$ such intervals. Therefore, there are at least $\lceil pK\rceil$ intervals $k$ such that $m_k \geq M - L/K$. Since $f < \lfloor pK \rfloor$, this implies that $m_f \geq M - L/K$. Similar arguments show that $m_f \leq M + L/K$.

We conclude by noting that since $m_f \geq M - L/K$, Lemma 2 implies $\min_{a \in \cup_{k \leq f} I_k} m(a) \geq M - 2L/K$. We define $\tilde{a} = \arg\max\{m(a) : a \in \cup_{k > f} \overline{I}_k\}$. The continuity of $m$ implies that $m(\tilde{a}) \geq M - 2L/K$. Let $\tilde{k} > f$ be such that $\tilde{a} \in \overline{I}_{\tilde{k}}$. Then, Lemma 2 implies that $m_{\tilde{k}} \geq M - 4L/K$. Since $m_{f+1} = \max_{k > f} m_k$, this implies in particular $m_{f+1} \geq M - 4L/K$. Similar arguments can be used to show that $m_{f+2} \geq M - 8L/K$ and that $m_{f-1} \leq M + 4L/K$. Thus, when $|\hat{f} - f| \leq 1$, we find that $m_{\hat{f}} \in [M - 4L/K, M + 4L/K]$, and $m_{\hat{f}+1} \in [M - 8L/K, M + L/K]$.

### D.3 Proof of Lemma 4

Recall that $\widehat{M} = m(a_{\phi^*(T)})$, where $T = pN$ and $\phi^*$ is a permutation such that $\{m(a_{\phi^*(i)})\}_{1 \leq i \leq N}$ is a decreasing sequence. Thus, $\widehat{M}$ is the $T$-th largest payment for the arms with covariates $\{a_1, ..., a_N\}$. To bound its deviation from its expected value $M$, we note that for all $t > 0$, $\left\{\widehat{M} \geq M + t\right\}$ implies

$$\left\{\sum_{1 \leq i \leq N} \mathbb{1}_{\{m(a_i) \geq M+t\}} \geq T\right\}. \text{ Since } T = Np = N\mathbb{P}\left(m(a_1) \geq M\right),$$

$$\mathbb{P}\left(\widehat{M} \geq M + t\right) \leq \mathbb{P}\left(\sum_{1 \leq i \leq N} \mathbb{1}_{\{m(a_i) \geq M+t\}} \geq N\mathbb{P}\left(m(a_1) \geq M\right)\right)$$

$$\leq \mathbb{P}\left(\sum_{1 \leq i \leq N} \left(\mathbb{1}_{\{m(a_i) \geq M+t\}} - \mathbb{P}\left(m(a_1) \geq M + t\right)\right) \geq N\mathbb{P}\left(m(a_1) \in [M, M+t)\right)\right).$$

Using Hoeffding's equality, we find that

$$\mathbb{P}\left(\widehat{M} \geq M + t\right) \leq \exp\left(-2N\mathbb{P}\left(m(a_1) \in [M, M+t)\right)^2\right).$$

For the choice $t = L/K$, it implies that

$$\mathbb{P}\left(\widehat{M} \geq M + L/K\right) \leq \exp\left(-2N\mathbb{P}\left(m(a_1) \in [M, M+L/K)\right)^2\right).$$

Next, we obtain a lower bound on $\mathbb{P}\left(m(a_1) \in [M, M + L/K)\right)$. Note that either $\max\{m(a) : a \in [0,1]\} \leq M + L/K$, and $\mathbb{P}\left(m(a_1) \in [M, M + L/K)\right) = \mathbb{P}\left(m(a_1) \geq M\right) = p \geq 1/K$, or $\max\{m(a) : a \in [0,1]\} > M + L/K$.

In this case, choose $a^{(1)} \in \arg\max_a\{m(a)\}$ ($a^{(1)}$ exists since $m$ is continuous and defined on a compact set). Note that $m(a^{(1)}) > M + L/K$. Since $m$ is continuous and $\lambda(\{a : m(a) < M\}) > 0$ (because of Assumption 3 and the fact that $p < 1$), $\{a : m(a) = M\} \neq \emptyset$. Define $a^{(2)} = \arg\min_a\{|a - a^{(1)}| : m(a) = M\}$, and assume without loss of generality that $a^{(1)} \leq a^{(2)}$. Since $m$ is continuous, $m(a^{(1)}) > M + L/K$ and $m(a^{(2)}) = M$, we see that $\{a \in [a^{(1)}, a^{(2)}] : m(a) = M + L/K\} \neq \emptyset$. Define finally $a^{(3)} = \max\{a : a \leq a^{(2)}, m(a) = M + L/K\}$. By construction, for all $a \in [a^{(3)}, a^{(2)})$, $m(a) \in [M, M + L/K)$. Using Assumption 2, we find that $|a^{(3)} - a^{(2)}| \geq 1/K$. Thus, $\mathbb{P}\left(m(a_1) \in [M, M + L/K)\right) \geq \mathbb{P}\left(m(a_1) \in [a^{(3)}, a^{(2)}]\right) \geq 1/K$.

Putting things together, we find that

$$\mathbb{P}\left(\widehat{M} \geq M + L/K\right) \quad \leq \quad \exp\left(-2N/K^2\right).$$

Using similar arguments, we can show that $\mathbb{P}\left(\widehat{M} \leq M - L/K\right) \leq \exp\left(-2N/K^2\right)$.

### D.4 Proof of Lemma 5

In order to prove Lemma 5, we first state the following result.

**Lemma 30.** *Let $\mathcal{B}$ be a Borel set of measure $\lambda(\mathcal{B}) \geq N^{-2/3}$, and $N_{\mathcal{B}}$ be the number of arms in $\mathcal{B}$. Then,*

$$\mathbb{P}\left(|N_{\mathcal{B}} - \lambda(\mathcal{B})N| \geq \sqrt{\lambda(\mathcal{B})N^{4/3}}\right) \leq 2e^{-\frac{N^{1/3}}{3}}.$$

*Proof.* Recall that $N_{\mathcal{B}} = \sum_{1 \leq i \leq N} \mathbb{1}_{a_i \in \mathcal{B}}$, where $\mathbb{1}_{a_i \in \mathcal{B}} \overset{i.i.d}{\sim} \text{Bernoulli}(\lambda(\mathcal{B}))$. Applying Bernstein's inequality, we find that

$$\mathbb{P}\left(|N_{\mathcal{B}} - \lambda(\mathcal{B})N| \geq t\right) \leq 2e^{-\frac{t^2}{2\lambda(\mathcal{B})N + 2t/3}}$$

$$\mathbb{P}\left(|N_{\mathcal{B}} - \lambda(\mathcal{B})N| \geq \sqrt{\lambda(\mathcal{B})N^{4/3}}\right) \leq 2e^{-\frac{N^{1/3}}{3}}$$

$\square$

Now, we use Lemma 30 for $\mathcal{B} = \{x : m(x) \in [M - 16L/K, M + L/K]\}$. By Assumption 3, $\lambda(\{x : m(x) \in [M - 16L/K, M + L/K]\}) \leq 16LQ/K$. When $K \leq N^{2/3}$, the inequality $QL \geq 1$ implies that $K \leq 16LQN^{2/3}$, and $\sqrt{16LQN^{4/3}}/K \leq 16LQN/K$. This proves Lemma 5.

### D.5   Proof of Lemma 6

Non-zero terms in $R_T^{(d)}$ correspond to pairs of arms $(i, j)$ such that $i$ is pulled by $\phi^d$ but not by $\phi^*$, and $j$ is pulled by $\phi^*$ but not by $\phi^d$. If an arm $i$ is pulled by $\phi^d$, it belongs to an interval $k$ such that $m_k \geq m_{\hat{f}+1}$. On the event $\mathcal{E}_a$,

$$m_{\hat{f}+1} \geq M - 8L/K.$$

Using Lemma 2, we find that

$$m(a_i) \geq M - 16L/K.$$

On the other hand, if $i$ is not pulled by $\phi^*$, it must be such that $m(a_i) \leq \widehat{M}$. On the event $\mathcal{E}_a$, this implies that $m(a_i) \leq M + L/K$. Since there are at most $\frac{32LQN}{K}$ arms in $[M - 16L/K, M + L/K]$ on the event $\mathcal{E}_a$, there are at most $\frac{32LQN}{K}$ arms that are selected by $\phi^d$ and not by $\phi^*$, and thus at most $\frac{32LQN}{K}$ non-zero terms in $R_T^{(d)}$.

Now, each of these terms corresponds to the cost of pulling an arm $i$ selected by $\phi^d$ but not by $\phi^*$, instead of an arm $j$ selected by $\phi^*$ but not by $\phi^d$. Assume that $m_{\hat{f}+1} \geq M$. Then, using Lemma 2, we see that if $i$ is selected by $\phi^d$, $m(a_i) \geq M - L/K$. Moreover, if $j$ is not selected by $\phi^d$, it belongs to an interval $I_k$ such that $m_k \leq m_{\hat{f}+1}$. On $\mathcal{E}_a$, $m_{\hat{f}+1} \leq M + L/K$. Thus, $m(a_j) \leq M + 2L/K$, and $m(a_j) - m(a_i) \leq 3L/K$. On the other hand, if $m_{\hat{f}+1} < M$, then according to Lemma 2 for all $i$ selected by $\phi^d$, $m(a_i) \geq M - 2\left((M - m_{\hat{f}+1}) \vee L/K\right)$, while for $j$ not selected by $\phi^d$, $m(a_j) \leq m_{\hat{f}+1} + (M - m_{\hat{f}+1})/2 \vee L/K$. Thus, $m(a_j) - m(a_i) \leq 3/2\left((M - m_{\hat{f}+1}) \vee 2L/K\right) \leq 12L/K$.

To conclude, on the event $\mathcal{E}_a$ there are at most $\frac{32LQN}{K}$ non-zero terms in $R_T^{(d)}$, and each of them is bounded by $12L/K$. Thus,

$$R_T^{(d)} \leq \frac{32QLN}{K} \times 12L/K.$$

### D.6   Proof of Lemma 7

Note that for $k \in \{1, ..., K\}$, $I_k$ is a Borel set of measure $1/K \geq N^{-2/3}$. Using Lemma 30, we find that

$$\mathbb{P}\left(\left|N_k - \frac{N}{K}\right| \geq \frac{N^{2/3}}{K^{1/2}}\right) \leq 2e^{-\frac{N^{1/3}}{3}}$$

$$\mathbb{P}\left(\left|N_k - \frac{N}{K}\right| \geq \frac{N}{2K} \times \frac{2K^{1/2}}{N^{1/3}}\right) \leq 2e^{-\frac{N^{1/3}}{3}}.$$

Since $K \leq N^{2/3}/4$, $2K^{1/2}/N^{1/3} \leq 1$. A union bound for $k = 1, ..., K$ yields the result.

### D.7 Proof of Lemma 8

Recall that $a_i \sim \mathcal{U}([0,1])$, and thus $a_i | \{a_i \in I_k\} \sim \mathcal{U}(I_k)$. Since the arms $a_{\pi_k(s)}$ are selected uniformly at random among the arms in $I_k$, they are independent from one another, and uniformly distributed on $I_k$.

For $k \in 1, ..., K$ and for $n \in [0, N]$, we denote by $\mathbb{P}_n$ the probability measure obtained by conditioning on the event $N_k = n$ (this event has a strictly positive probability because $\lambda(I_k) \in (0,1)$). Note that for any $s \in [1, n]$, $\mathbb{E}_n[y_{\pi_k(s)}] = m_k$. Using Hoeffding's inequality, we find that for any $n \in [1, N]$ and any $s \in [1, n]$

$$\mathbb{P}_n \left( \left| \frac{1}{s} \sum_{1 \leq i \leq s} y_{\pi_k(i)} - m_k \right| \geq \sqrt{\frac{\log(T/\delta)}{2s}} \right) \leq 2e^{-\log(T/\delta)} = \frac{2\delta}{T}$$

The inequality $|\widehat{m}_k(0) - m_k| \leq \infty$ also holds, since we defined $\widehat{m}_k(0) = 0$, and thus the inequality above is also verified for $n = 0$. Using a union bound for $s = 0, ..., (n \wedge T)$, we find that for all $n = 0, ..., N$,

$$\mathbb{P}_n \left( \exists s \leq (n \wedge T) : |\widehat{m}_k(s) - m_k| \geq \sqrt{\frac{\log(T/\delta)}{2s}} \right) \leq \frac{2\delta(n \wedge T)}{T} \leq 2\delta.$$

We integrate over the different values of $n$ and find that

$$\mathbb{P} \left( \exists s \leq (N_k \wedge T) : |\widehat{m}_k(s) - m_k| \geq \sqrt{\frac{\log(T/\delta)}{2s}} \right) \leq 2\delta.$$

Finally, a union bound for $k = 1, ..., K$ yields

$$\mathbb{P} \left( \exists k \in \{1, ..., K\}, s \leq (N_k \wedge T) : |\widehat{m}_k(s) - m_k| \geq \sqrt{\frac{\log(T/\delta)}{2s}} \right) \leq 2K\delta.$$

### D.8 Proof of Lemma 9

First, note that on $\mathcal{E}_b$, all intervals are non-empty. By definition of Algorithm UCBF, at least one arm is pulled in each interval. To bound the number of arms pulled in interval $I_l$, assume that time $t > K$ is such that the arm $\phi(t)$ is selected in $I_l$. Since there are arms available in $I_k$ at time $T$, there are arms available in $I_k$ at time $t \leq T$. If UCBF pulls an arm in $I_l$ instead of an arm in $I_k$, we must have

$$\widehat{m}_k(n_k(t-1)) + \sqrt{\frac{\log(T/\delta)}{2n_k(t-1)}} \leq \widehat{m}_l(n_l(t-1)) + \sqrt{\frac{\log(T/\delta)}{2n_l(t-1)}}.$$

On the event $\mathcal{E}_b$, this implies that

$$m_k \leq m_l + 2\sqrt{\frac{\log(T/\delta)}{2n_l(t-1)}}.$$

Straightforward calculations show that

$$n_l(t-1) \leq \frac{2\log(T/\delta)}{\Delta_{k,l}^2}.$$

Thus $n_l(T) \leq \left( \frac{2\log(T/\delta)}{\Delta_{k,l}^2} \vee 1 \right) + 1 \leq \frac{3\log(T/\delta)}{\Delta_{k,l}^2}$ since $\Delta_{k,l}^2 \leq 1$ and $\log(T/\delta) \geq 1$.

### D.9 Proof of Lemma 10

By Lemma 3, on the event $\mathcal{E}_a$, $m_{\widehat{f}+1} \in [M - 8L/K, M + L/K]$. We group intervals with mean rewards lower than $m_{\widehat{f}+1}$ into the following subsets.

Let $\mathcal{S}_0 = \{k : (M - m_k) \in [-L/K, 10L/K]\}$, $\mathcal{S}_1 = \{k : (M - m_k) \in (10L/K, 16L/K]\}$, and for $n \geq 2$ define $\mathcal{S}_n = \{k : (M - m_k) \in [2^{n+2}L/K, 2^{n+3}L/K\}$. Note that for $n \geq \log_2(K/L) - 2$, $\mathcal{S}_n$ is empty since $m$ is bounded by 1.

Using Lemma 2, we note that for all $l \in \mathcal{S}_0$ and all $a \in I_l$,
$$|m(a) - M| \leq 20L/K.$$
Using Assumption 3, we conclude that $|\mathcal{S}_0| \leq 20LQ$. On $\mathcal{E}_a$, there are at most $1.5N/K$ arms in each interval, so the number of arms in intervals in $\mathcal{S}_0$ is at most $30LQN/K$. Moreover for all $l \in \mathcal{S}_0$ and all $a_i \in I_l$, $(M - m(a_i)) \leq 20L/K$. Thus, the arms pulled from intervals in $\mathcal{S}_0$ contributes to $R_{subopt}$ by at most $20L/K \times 30LQN/K = 600L^2QN/K^2$.

Similarly, for all $l \in \mathcal{S}_1$ and all $a \in I_l$,
$$|m(a) - M| \leq 32L/K.$$
Using Assumption 3, we conclude that $|\mathcal{S}_1| \leq 32LQ$. Moreover, by definition of $\widehat{f}$, there exists an interval $I_k$ with $m_k \geq m_{\widehat{f}+1}$ such that $n_k(T) < N_k$. Since $\Delta_{k,l} \geq m_{\widehat{f}+1} - m_l \geq M - 8L/K - (M - 10L/K) \geq 2L/K$ for all $l \in \mathcal{S}_1$, we use Lemma 9 and find that
$$n_l(T) \leq \frac{3\log(T/\delta)K^2}{4L^2}.$$
Thus, the number of arms pulled in $\mathcal{S}_1$ is at most $3\log(T/\delta)K^2/(4L^2) \times 32LQ = 24\log(T/\delta)K^2Q/L$. Since each arm in $\mathcal{S}_1$ has a payment larger than $M - 32L/K$, the arms pulled from intervals in $\mathcal{S}_1$ contributes to $R_{subopt}$ by at most $24\log(T/\delta)K^2Q/L \times 32L/K \leq 768\log(T/\delta)KQ$.

Finally, note that for $n \geq 2$ and $l \in \mathcal{S}_n$, $\Delta_{k,l} \geq (2^{n+2} - 8)L/K \geq 2^{n+1}L/K$. Using Lemma 9, we find that
$$n_l(T) \leq \frac{3\log(T/\delta)K^2}{2^{2n+2}L^2}.$$
Applying Lemma 2, we see that each arm $a_i \in I_l$ verifies $m(a_i) \geq M - 2^{n+4}L/K$. Using Assumption 3, we find that $|\mathcal{S}_n| \leq 2^{n+4}QL$. Thus,

$$
\begin{aligned}
R_{subopt} &\leq \frac{600L^2QN}{K^2} + 768\log(T/\delta)KQ \\
&\quad + \sum_{n=2}^{\log_2(K/L)-2} |\mathcal{S}_n| \frac{3\log(T/\delta)K^2}{2^{2n+2}L^2} \times \frac{2^{n+4}L}{K} \\
&\leq \frac{600L^2QN}{K^2} + 768\log(T/\delta)KQ + 192(\log(T/\delta)QK(\log_2(K/L) - 3)) \\
&\leq \frac{600L^2QN}{K^2} + 192\log(T/\delta)KQ\left(\log_2(K/L) + 1\right).
\end{aligned}
$$

### D.10 Proof of Lemma 11

Along the lines of the proof of Lemma 10, we have proved that on $\mathcal{E}_a \cap \mathcal{E}_b$, the number of arms in $\mathcal{S}_0$ is bounded by $30QLN/K$ and that the number of arms pulled from intervals in $\mathcal{S}_1$ is bounded by $24\log(T/\delta)K^2Q/L$. Thus,

$$
\begin{aligned}
\sum_{n=0}^{\log_2(K/L)-1} \sum_{I_k \in \mathcal{S}_n} n_k(T) &\leq \frac{30QLN}{K} + \frac{24\log(T/\delta)K^2Q}{L} + \sum_{n=2}^{\log_2(K/L)-1} |\mathcal{S}_n| \frac{3\log(T/\delta)K^2}{2^{2n+2}L^2} \\
&\leq \frac{30QLN}{K} + \frac{24\log(T/\delta)K^2Q}{L} + \sum_{n=2}^{\log_2(K/L)-1} \frac{12\log(T/\delta)QK^2}{2^nL} \\
&\leq \frac{30QLN}{K} + \frac{30\log(T/\delta)K^2Q}{L}
\end{aligned}
$$

Thus, the number of arms pulled from sub-optimal intervals is bounded by $30Q(LN/K + \log(T/\delta)K^2/L)$.

### D.11 Proof of Lemma 12

Before proving Lemma 12, let us introduce further notations. For any two intervals $I_h$ and $I_i$ such that $m_h \geq m_i$, define $N_{[h,i]} = \sum_{j=h}^{i} N_j$, and $n_{[h,i]}(T) = \sum_{j=h}^{i} n_j(T)$.

We prove Lemma 12 by contradiction. Assume that there is an interval $I_k$ such that $m_k \geq M+AL/K$ and $n_k(T) < N_k$. By continuity of $m$, there exists $a \in [0,1]$ such that $m(a) = M + AL/(4K)$, and by Lemma 2 there exists an interval $I_l$ that contains $a$ such that $m_l \in [M+AL/(8K), M+AL/(2K)]$. Note that since $A \geq 33$, on the event $\mathcal{E}_a$ $m_l > m_{\hat{f}}$ and $l < \hat{f}$.

By definition of $\hat{f}$, we have $T > N_{[1,\hat{f}]} = N_{[1,l-1]} + N_{[l,\hat{f}]}$. On the other hand, $T = n_{[1,l-1]}(T) + n_{[l,K]}(T)$. Since $N_{[1,l-1]} > n_{[1,l-1]}(T)$ on the event $\{n_k(T) < N_k\}$, we necessarily have $N_{[l,\hat{f}]} < n_{[l,K]}(T) = n_{[l,\hat{f}]}(T) + n_{[\hat{f}+1,K]}(T)$.

We obtain a contradiction by proving that on $\mathcal{E}_a \cap \mathcal{E}_b \cap \{n_k(T) < N_k\}$,

$$N_{[l,\hat{f}]} - n_{[l,\hat{f}]}(T) > n_{[\hat{f}+1,K]}(T).$$

In words, we prove that the number of sub-optimal arms pulled is strictly smaller than the number of remaining optimal arms, and obtain a contradiction.

To obtain a lower bound on $N_{[l,\hat{f}]} - n_{[l,\hat{f}]}(T)$, we note that for all $h \in [l, \hat{f}]$, $\Delta_{k,h} \geq \Delta_{k,l} \geq AL/(2K)$. Using Lemma 9, we see that of the event $\mathcal{E}_a \cap \mathcal{E}_b \cap \{n_k(T) < N_k\}$

$$n_h(T) \leq \frac{3\log(T/\delta)}{\Delta_{k,h}^2} \leq \frac{3\log(T/\delta)}{(AL/(2K))^2} \leq \frac{12K^2\log(T/\delta)}{(AL)^2}.$$

On the event $\mathcal{E}_b$, each interval contains at least $N/(2K)$ arms. Thus,

$$N_h - n_h(T) \geq \frac{N}{2K} - \frac{12K^2\log(T/\delta)}{(AL)^2}.$$

Let $\mathcal{N}_{[l,\hat{f}]}$ denote the number of intervals $I_h$ for $h \in [l, \hat{f}]$, and let $a^{(1)} \in I_l$ be such that $m(a^{(1)}) = M+AL/(4K)$. Let $a^{(2)} = \arg\min_{a:m(a)=M+4L/K}|a-a^{(1)}|$, and assume without loss of generality that $a^{(1)} < a^{(2)}$. Let $a^{(3)} = max\{a \in [a^{(1)}, a^{(2)}] : m(a) = M + AL/(4K)\}$. All interval $h$ such that $I_h \subset [a^{(3)}, a^{(2)}]$ have mean reward in $[M + 4L/K, M + AL/(4K)]$. On the event $\mathcal{E}_a$, those intervals belong to $[l, \hat{f}]$. Using Assumption 2, we find that $L|a^{(2)}-a^{(3)}| \vee 4L/K \geq (A-16)L/(4K)$, and so $|a^{(2)} - a^{(3)}| \geq (A-16)/(4K) \geq A/(8K)$ (since $A > 32$). The number of intervals of size $1/K$ in $[a^{(3)}, a^{(2)}]$ is therefore at least $A/8 - 1$. Thus $\mathcal{N}_{[l,\hat{f}]} \geq A/8 - 1$, and

$$N_{[l,\hat{f}]} - n_{[l,\hat{f}]}(T) \geq \left(\frac{A}{8} - 1\right)\left(\frac{N}{2K} - \frac{12K^2\log(T/\delta)}{(AL)^2}\right).$$

Since $A > 32$, $A/8 - 1 \geq 3A/32$. Thus

$$N_{[l,\hat{f}]} - n_{[l,\hat{f}]}(T) \geq \frac{3A}{32}\left(\frac{N}{2K} - \frac{12K^2\log(T/\delta)}{(AL)^2}\right).$$

To obtain an upper bound on $n_{[\hat{f}+1,K]}(T)$, we divide the intervals $\hat{f} + 1, ..., K$ into subsets. Let $\widetilde{\mathcal{S}}_0 = \{l : M - m_l \in [-4L/K, AL/K]\}$, and for $n > 0$ let $\widetilde{\mathcal{S}}_n = \{l : M - m_l \in [AL/K \times 2^{n-1}, AL/K \times 2^n]\}$. Since $m_{\hat{f}} \leq M + 4L/K$, we see that $\{\hat{f} + 1, ..., K\} \subset \bigcup_{n \geq 0} \widetilde{\mathcal{S}}_n$.

For all $h \in \widetilde{\mathcal{S}}_0$, $\Delta_{k,h} \geq (A-4)L/K \geq 7AL/(8K)$ since $A > 32$. Similarly, for all $n > 0$ and all $h \in \widetilde{\mathcal{S}}_n$, $\Delta_{k,h} \geq AL/(K(1+2^{n-1})) \geq AL/(2^{n-1}K)$. Using Lemma 9, we find that on the event $\mathcal{E}_a \cap \mathcal{E}_b \cap \{n_k(T) < N_k\}$,

$$
n_{[\hat{f}+1,K]}(T) \quad \leq \quad |\widetilde{\mathcal{S}}_0| \frac{192K^2 \log(T/\delta)}{49A^2L^2} + \sum_{n \geq 1} |\widetilde{\mathcal{S}}_n| \frac{3 \log(T/\delta)}{\left(AL/K\right)^2 2^{2n-2}}
$$

Using Lemma 2 and Assumption 3, we find that $|\widetilde{\mathcal{S}}_0| \leq 2ALQ$ and that for any $n > 0$, $|\widetilde{\mathcal{S}}_n| \leq 2^{n+1}ALQ$. This implies that

$$
\begin{aligned}
n_{[\hat{f}+1,K]}(T) \quad &\leq \quad \frac{384QK^2 \log(T/\delta)}{49AL} + \sum_{n \geq 1} \frac{48 \log(T/\delta)QK^2}{AL2^n} \\
&\leq \quad \frac{48K^2 \log(T/\delta)Q}{AL} \left( \frac{8}{49} + \sum_{n \geq 1} \frac{1}{2^n} \right) \\
&\leq \quad \frac{56K^2 \log(T/\delta)Q}{AL}
\end{aligned}
$$

Note that we necessarily have $QL \geq 1$. Thus, for the choice $A = 35\sqrt{\frac{K^3Q \log(T/\delta)}{NL} \vee 1}$, we find that $N_{[l,\hat{f}]} - n_{[l,\hat{f}]}(T) > n_{[\hat{f}+1,K]}(T)$, which is impossible. We conclude that all intervals $I_h$ with a mean reward larger than $AL/K$ have been killed.

### D.12 Proof of Lemma 13

We have shown in Lemma 11 that the number of non-zero terms in $R_{opt}$ is bounded by $30Q(LN/K + \log(T/\delta)K^2/L)$. Moreover, in Lemma 12, we have shown that those non-zero terms correspond to arms $a_i$ in intervals $I_k$ such that $m_k \leq M + AL/K$. By Assumption 2, their payments $m(a_i)$ are such that $m(a_i) \leq M + 2AL/K$, so each non zero term is bounded by $2AL/K$. Thus, we find that

$$
R_{opt} \leq 30Q \left( \frac{LN}{K} + \frac{\log(T/\delta)K^2}{L} \right) \times 2AL/K
$$

### D.13 Proof of Lemma 14

The functions $m_0$ and $m_1$ are piecewise linear with slopes $\tilde{L}$ and $-\tilde{L}$. Since $\tilde{L} = L \wedge 1/2 \leq L$, Assumption 2 is satisfied.

On the other hand, for $\epsilon \in (0, \tilde{L}\delta)$,

$$
\begin{aligned}
\lambda \left( \{x : |m_0(x) - 0.5| \leq \epsilon\} \right) \quad &= \quad \lambda \left( [x_0 - \epsilon/\tilde{L}, x_0 + \epsilon/\tilde{L}] \right) + \lambda \left( [1 - p - \epsilon/\tilde{L}, 1 - p + \epsilon/\tilde{L}] \right) \\
&\quad + \lambda \left( [x_1 - \epsilon/\tilde{L}, x_1 + \epsilon/\tilde{L}] \right) \\
&= \quad 6\epsilon/\tilde{L} = 6\epsilon \times (1/L \vee 2) \leq Q\epsilon.
\end{aligned}
$$

For $\epsilon \geq \tilde{L}\delta$,

$$
\begin{aligned}
\lambda \left( \{x : |m_0(x) - 0.5| \leq \epsilon\} \right) \quad &= \quad \lambda \left( [x_0 - \epsilon/\tilde{L}, x_1 + \epsilon/\tilde{L}] \right) = x_1 - x_0 + 2\epsilon/\tilde{L} \\
&= \quad 4\delta + 2\epsilon/\tilde{L} \leq 6\epsilon \times (1/L \vee 2) \leq Q\epsilon.
\end{aligned}
$$

Thus, $m_0$ satisfies Assumption 3. The same holds for $m_1$.

### D.14 Proof of Lemma 15

Recall that $\Phi(T) = \{\phi(1), ..., \phi(T)\}$. We bound the Kullback-Leibler divergence between $\mathbb{P}_0$ and $\mathbb{P}_1$ (see, e.g., Lemma 15.1 in [Lattimore and Szepesvári, 2020]):

$$
\begin{aligned}
KL(\mathbb{P}_0, \mathbb{P}_1) &= \sum_{i=1,...,N} \mathbb{E}_0 \left[ \mathbb{1}_{i \in \Phi(T)} \right] KL(\mathcal{P}_0^{y_i}, \mathcal{P}_1^{y_i}) \\
&\leq \sum_{i=1,...,N} KL(\mathcal{P}_0^{y_i}, \mathcal{P}_1^{y_i})
\end{aligned}
$$

where $KL(\mathcal{P}_0^{y_i}, \mathcal{P}_1^{y_i})$ denotes the Kullback-Leibler divergence of the distribution of the reward $y_i$ under $m_0$ and $m_1$. For $p, q \in (0, 1)$, we denote by $kl(p, q)$ the Kullback-Leibler divergence between two Bernoulli of means $p$ and $q$. Since the variables $y_i$ are Bernoulli random variable of parameter $m(a_i)$, we find that

$$
\begin{aligned}
KL(\mathbb{P}_0, \mathbb{P}_1) &\leq \sum_{i=1,...,N} kl(m_0(a_i), m_1(a_i)) \\
&\leq \sum_{a_i \in [x_0, x_1]} kl(m_0(a_i), m_1(a_i)).
\end{aligned}
$$

By definition of $m_0$ and $m_1$, for all $a_i \in [x_0, x_1]$, $|0.5 - m_0(a_i)| = |0.5 - m_1(a_i)| \leq \delta \tilde{L} < 1/4$. Easy calculations show that for $\epsilon \in [-1/2, 1/2]$,

$$
kl \left( \frac{1 - \epsilon}{2}, \frac{1 + \epsilon}{2} \right) \leq 4\epsilon^2.
$$

Using Assumption 5 and the definition of $m_0$ and $m_1$, we find that

$$
\begin{aligned}
KL(\mathbb{P}_0, \mathbb{P}_1) &\leq 4 \sum_{i=0}^{\lceil N\delta \rceil} 4 \left( 2\tilde{L} \frac{i}{N} \right)^2 \\
&\leq \frac{64\tilde{L}^2}{N^2} \times (N\delta + 1)^3.
\end{aligned}
$$

Now, since $\alpha \geq 20 N^{-2/3}$ and $\tilde{L}^{-2/3} \geq 0.5^{-2/3}$, $N\delta = N^{2/3} \alpha \tilde{L}^{-2/3} \geq 31$, and thus $(N\delta + 1)^3 \leq (N\delta)^3 (1 + 1/31)^3 \leq 1.1 (N\delta)^3$. Thus,

$$
KL(\mathbb{P}_0, \mathbb{P}_1) \leq \frac{70.4\tilde{L}^2}{N^2} \times (N\delta)^3 \leq 70.4\alpha^3.
$$

### D.15 Proof of Lemma 16

Under $\mathbb{P}_0$, we can see that all arms in $(x_0, 1 - p)$ are sub-optimal. By construction, all optimal arms have a payment higher than $1/2$. Thus,

$$
R_T \geq \sum_{a_i \in (x_0, 1-p)} \left( \frac{1}{2} - m_0(a_i) \right) \mathbb{1}\{i \in \Phi(T)\}.
$$

There are at least $\lfloor N\delta \rfloor$ arms in $[x_0, x_0 + \delta)$, and at least $\lfloor N\delta \rfloor$ arms in $[x_0 + \delta, 1 - p]$. We use the change of variables $k = i - \lceil x_0 N \rceil$ and $k = \lceil (1-p)N \rceil - i$ to sum over the indices of the sub-optimal arms. We find that

$$
\begin{aligned}
R_T \geq \quad & \sum_{k=0}^{\lfloor N\delta \rfloor - 1} \frac{k\tilde{L}}{N} \mathbb{1}\{(\lceil x_0 N \rceil + k) \in \Phi(T)\} \\
& + \sum_{k=0}^{\lfloor N\delta \rfloor} \frac{k\tilde{L}}{N} \mathbb{1}\{((1-p)N - k) \in \Phi(T)\}.
\end{aligned}
$$

On $\mathcal{Z}$, at least $\lfloor N\delta \rfloor - 2$ arms are pulled in $(x_0, 1-p)$, so easy calculations lead to

$$R_T \;\geq\; 2\sum_{k=0}^{\lfloor 0.5N\delta \rfloor - 2}\frac{k\tilde{L}}{N}$$

$$\geq\; \frac{2\tilde{L}}{N}\frac{(\lfloor 0.5N\delta \rfloor - 1)(\lfloor 0.5N\delta \rfloor - 2)}{2}.$$

We have shown in Lemma 15 that $N\delta \geq 31$, so $(\lfloor 0.5N\delta \rfloor - 1)(\lfloor 0.5N\delta \rfloor - 2) \geq 2^{-2}(N\delta)^2(1 - 4/31)(1 - 6/31)$. Thus,

$$R_T \;\geq\; 2^{-2}(1-4/31)(1-8/31)\frac{(N\delta)^2\tilde{L}}{N} \geq 2^{-2}(1-4/31)(1-6/31)\tilde{L}^{-1/3}\alpha^2 N^{1/3}.$$

Since $\tilde{L} \leq 1/2$, this implies

$$R_T \;\geq\; 0.22\alpha^2 N^{1/3}.$$

On the other hand, all arms in $(x_0, 1-p)$ are optimal for the payoff function $m_1$. Since all sub-optimal arms have a payment at most $1/2$, under $\mathcal{P}_1$,

$$R_T \;\geq\; \sum_{a_i \in [x_0, 1-p]}\left(m_1(a_i) - \frac{1}{2}\right)\mathbb{1}\{i \notin \Phi(T)\}.$$

Applying the argument developed previously, we find that

$$R_T \;\geq\; \sum_{k=0}^{\lfloor N\delta \rfloor - 1}\frac{k\tilde{L}}{N}\mathbb{1}\left\{(k + \lceil x_0 N \rceil) \in \Phi(T)\right\}$$

$$+ \sum_{k=0}^{\lfloor N\delta \rfloor}\frac{k\tilde{L}}{N}\mathbb{1}\left\{((1-p)N - k) \in \Phi(T)\right\}.$$

On $\overline{\mathcal{Z}}$, at most $\lfloor N\delta \rfloor - 2$ arms are pulled in $(x_0, 1-p)$. Under Assumption 5, there are at least $\lfloor 2N\delta \rfloor - 2$ arms in $(x_0, 1-p)$. All of these arms are optimal for the payoff function $m_1$. Thus, on $\overline{\mathcal{Z}}$, the number of sub-optimal arms pulled is at least $\lfloor 2N\delta \rfloor - 2 - (\lfloor N\delta \rfloor - 2) \geq \lfloor N\delta \rfloor$. Thus,

$$R_T \;\geq\; 2\sum_{k=0}^{\lfloor 0.5N\delta \rfloor - 1}\frac{k\tilde{L}}{N} \geq \frac{\tilde{L}}{N}(0.5N\delta - 2)(0.5N\delta - 1) \geq \frac{2^{-2}(N\delta)^2\tilde{L}}{N}(1-4/31)(1-2/31).$$

We use $\tilde{L} \leq 1/2$ to find that $R_T \geq 0.25\alpha^2 N^{1/3}$.

### D.16 Proof of Lemma 18

By definition of $M$,

$$\mathbb{P}\left(m(a_1) \in [M, M+t)\right) \;=\; \mathbb{P}\left(m(a_1) \geq M\right) - \mathbb{P}\left(m(a_1) \geq M+t\right)$$

$$=\; p - \mathbb{P}\left(m(a_1) \geq M+t\right).$$

To provide an upper bound on $\mathbb{P}\left(m(a_1) \geq M+t\right)$, we use gaussian isoperimetric inequalities (see, e.g., Chapter 5.1 in [Vershynin, 2018]). Those results can readily be extended to random variable uniformly distributed on the unit cube. To do so, we introduce a random normal variable $z = (z_1, ..., z_d) \sim \mathcal{N}(0, I_d)$, and we denote by $F$ the c.d.f. of a $z_1$. Moreover, we introduce a new payment function

$$\tilde{m} : (z_1, ..., z_d) \to m(F(z_1), ..., F(z_d)).$$

It is easy to see that $\tilde{m}(z)$ and $m(a_1)$ have the same distribution. Thus, by definition of $p$,

$$\mathbb{P}\left(m(a_1) \in [M, M+t)\right) = p - \mathbb{P}\left(\tilde{m}(z) \geq M+t\right). \tag{11}$$

Next, we show that $\tilde{m}$ verifies a weak Lipschitz Assumption. Indeed, for any $z = (z_1, ..., z_d) \in \mathbb{R}^d$, and $z' = (z_1', ..., z_d') \in \mathbb{R}^d$, by definition of $\tilde{m}$

$$
\begin{aligned}
|\tilde{m}(z) - \tilde{m}(z')| &= |m\left(F(z_1), ..., F(z_d)\right) - m\left(F(z_1'), ..., F(z_d')\right)| \\
&\leq |M - \tilde{m}(z)| \vee L \left\| (F(z_1), ..., F(z_d)) - (F(z_1'), ..., F(z_d')) \right\|_2
\end{aligned}
$$

where the last equation follows from Assumption 2. Now, the gaussian c.d.f. $F$ is Lipschitz continuous, with Lipschitz constant equal to $(2\pi)^{-1/2}$. Thus,

$$
|\tilde{m}(z) - \tilde{m}(z')| \leq |M - \tilde{m}(z)| \vee (L \times (2\pi)^{-1/2} \|z - z'\|_2)
$$

Thus, for all $z \in \mathbb{R}^d$ such that $\tilde{m}(z) \geq M + t$ and all $z' \in \mathbb{R}^d$ such that $\tilde{m}(z') < M$, necessarily $\|z - z'\|_2 \geq \sqrt{2\pi} t/L$.

Let us denote by $\mathcal{B}$ the set of Borel sets of $\mathbb{R}^d$, and by $d(z, A)$ the Euclidean distance between a point $z \in \mathbb{R}^d$ and a set $A \in \mathcal{B}$. Moreover, let us denote by $A = \{z \in \mathbb{R}^d : \tilde{m}(z) < M\}$ the sub-level set of level $M$ of the function $\tilde{m}$. By definition of $M$, we have $\mathbb{P}(A) \leq 1 - p$. Moreover, the results above show that $\{z \in \mathbb{R}^d : \tilde{m}(z) \geq M + t\} \subset \{z \in \mathbb{R}^d : d(z, A) \geq \sqrt{2\pi} t/L\}$. This implies that

$$
\begin{aligned}
\mathbb{P}\left(\tilde{m}(z) \geq M + t\right) &\leq \mathbb{P}\left(d(z, A) \geq \sqrt{2\pi} t/L\right) \\
&\leq \sup_{B \in \mathcal{B}: \mathbb{P}(B) \leq 1-p} \mathbb{P}\left(d(z, B) \geq \sqrt{2\pi} t/L\right).
\end{aligned}
\tag{12}
$$

By Theorem 5.2.1 in [Vershynin, 2018], $\mathbb{P}\left(d(z, B) \geq \sqrt{2\pi} t/L\right)$ is maximized under the constraint $\mathbb{P}(B) \leq 1 - p$ when $B$ is a half space of gaussian measure $1 - p$. This is the case, for example, when $B = \{x \in \mathbb{R}^d : \langle x | e_1 \rangle \geq F^{-1}(p)\}$ and $e_1 = (1, 0, ..., 0)$ is the first vector of the canonical basis of $\mathcal{R}^d$. Then,

$$
\left\{z : d(Z, B) \geq \sqrt{2\pi} t/L\right\} = \left\{z = (z_1, ..., z_d) : z_1 \leq F^{-1}(p) - \sqrt{2\pi} t/L\right\}.
$$

Then, Equation (12) implies

$$
\begin{aligned}
\mathbb{P}\left(\tilde{m}(z) \geq M + t\right) &\leq P\left(z_1 \leq F^{-1}(p) - \sqrt{2\pi} t/L\right) \\
&= F\left(F^{-1}(p) - \sqrt{2\pi} t/L\right).
\end{aligned}
\tag{13}
$$

Combining Equations (11) and (13), we find that

$$
\begin{aligned}
\mathbb{P}\left(m(a_1) \in [M, M+t)\right) &\geq p - F\left(F^{-1}(p) - \sqrt{2\pi} t/L\right) \\
&= F(F^{-1}(p)) - F\left(F^{-1}(p) - \sqrt{2\pi} t/L\right).
\end{aligned}
$$

Using the c.d.f. of the normal distribution, we find that

$$
\mathbb{P}\left(m(a_1) \in [M, M+t)\right) \geq \int_{F^{-1}(p) - \sqrt{2\pi} t/L}^{F^{-1}(p)} \frac{1}{\sqrt{2\pi}} e^{\frac{-z^2}{2}} dz
\tag{14}
$$

$$
\geq \frac{t}{L} e^{\frac{-(F^{-1}(p) - \sqrt{2\pi} t/L)^2}{2}}.
\tag{15}
$$

We recall that $t/L \leq \sqrt{d}$, and conclude that

$$
\mathbb{P}\left(m(a_1) \in [M, M+t)\right) \geq \frac{t}{L} e^{-(F^{-1}(p) - \sqrt{2\pi d})^2/2}.
$$

### D.17 Proof of Lemma 19

Recall that $\epsilon = \lceil K^{d-1} c_{p,d} \rceil$. Similarly to the one-dimensional case, we begin by proving that $\hat{f} \geq f - \epsilon$. Since this inequality becomes trivial if $\epsilon \geq f$, we assume that $\epsilon < f$. Recall that $f = \lfloor pK^d \rfloor$, and $\hat{f}$ is such that $N_1 + .. + N_{\hat{f}} < T \leq N_1 + .. + N_{\hat{f}+1}$. By definition,

$N_1 + .. + N_{f-\epsilon} = \sum\limits_{1 \leq i \leq N} \mathbb{1}_{\{a_i \in B_1 \cup .. \cup B_{f-\epsilon}\}}$, where $\mathbb{1}_{\{a_i \in B_1 \cup .. \cup B_{f-\epsilon}\}}$ are independent Bernoulli random variables of parameter $\frac{f-\epsilon}{K^d}$. Using Hoeffding's inequality, we see that for all $t > 0$,

$$\mathbb{P}\left(\sum\limits_{1 \leq i \leq N} \mathbb{1}_{\{a_i \in B_1 \cup .. \cup B_{f-\epsilon}\}} - \frac{(f-\epsilon)N}{K^d} \geq t\right) \leq \exp\left(-\frac{2t^2}{N}\right).$$

Choosing $t = \epsilon N/K^d \geq c_{p,d}N/K$, we see that

$$\mathbb{P}\left(\sum\limits_{1 \leq i \leq N} \mathbb{1}_{\{a_i \in B_1 \cup .. \cup B_{f-\epsilon}\}} - \frac{(f-\epsilon)N}{K^d} \geq \frac{\epsilon N}{K^d}\right) \leq \exp\left(-\frac{2c_{p,d}^2 N}{K^2}\right).$$

Now, by definition, $f = \lfloor TK^d/N \rfloor$, and so $fN/K^d \leq T$. Thus,

$$\mathbb{P}\left(N_1 + .. + N_{f-\epsilon} \geq T\right) \leq \exp\left(-\frac{2c_{p,d}^2 N}{K^2}\right). \tag{16}$$

This shows that with high probability, $N_1 + .. + N_{f-\epsilon} < T$, which implies that $f - \epsilon < \hat{f} + 1$. Similarly, we can show that with probability at least $1 - \exp\left(-\frac{2c_{p,d}^2 N}{K^2}\right)$, $f + \epsilon + 1 \geq \hat{f}$. Thus, with probability larger than $1 - 2\exp\left(-\frac{2c_{p,d}^2 N}{K^2}\right)$, $|f - \hat{f}| \leq 1 + \epsilon$.

In a second time, we prove that $m_f \in [M - L\sqrt{d}/K, M + L\sqrt{d}/K]$.

We first show that there are at least $\lceil pK^d \rceil$ bins $k$ such that $m_k \geq M - L\sqrt{d}/K$, or equivalently that there are at most $\lfloor (1-p)K^d \rfloor$ bins $k$ such that $m_k < M - L\sqrt{d}/K$. Indeed, for all $k$ such that $m_k < M - L\sqrt{d}/K$, there exists $a \in B_k$ such that $m(a) < M - L\sqrt{d}/K$. Using Lemma 2, we see that $\forall a \in B_k$, $m(a) \leq M$. By definition of $M$, there can be at most $\lfloor (1-p)K^d \rfloor$ such bins. Therefore, there are at least $\lceil pK^d \rceil$ bins $k$ such that $m_k \geq M - L\sqrt{d}/K$. Since $f < \lfloor pK^d \rfloor$, this implies that $m_f \geq M - L\sqrt{d}/K$. Similar arguments show that $m_f \leq M + L\sqrt{d}/K$.

Now, recall that $\alpha = 4QL/c_{p,d} + 2/\sqrt{d} \times (1 + 3/K^{d-1})$. We show that $m_{f-\epsilon-2} \leq M + \alpha L\sqrt{d}/K$. Note that by Assumption 2 and by definition of $M$, $\max_a\{m(a)\} \leq M + L\sqrt{d}$. Then, if $\alpha/K \geq 1$, $m_{f-\epsilon-2} \leq M + \alpha L\sqrt{d}/K$ is automatically verified. We therefore restrict our attention to the case $\alpha/K < 1$. Now, we show that there are at least $\epsilon + 2$ bins $B_k$ such that $m_k \in [m_f, M + \alpha L\sqrt{d}/K]$. Applying Lemma 18 and Assumption 3, we find that

$$\lambda\left(\left\{a : m(a) \in [M + 2L\sqrt{d}/K, M + \alpha L\sqrt{d}/(2K)]\right\}\right)$$
$$= \lambda\left(\left\{a : m(a) \in [M, M + \alpha L\sqrt{d}/(2K)]\right\}\right) - \lambda\left(\left\{a : m(a) \in [M, M + 2L\sqrt{d}/K]\right\}\right)$$
$$\geq \alpha c_{p,d}\sqrt{d}/(2K) - 2QL\sqrt{d}/K = c_{p,d}(1 + 3/K^{d-1}).$$

Using Lemma 17, we see that all arms $a$ such that $m(a) \in [M + 2L\sqrt{d}/K, M + \alpha L\sqrt{d}/(2K)]$ belongs to bins $B_k$ such that $m_k \in [M + L\sqrt{d}/K, M + \alpha L\sqrt{d}/K]$. Thus, the number of bins with mean reward in $[M + L\sqrt{d}/K, M + \alpha L\sqrt{d}/K]$ is at least $c_{p,d}\left(1 + 3/K^{d-1}\right) \times K^d$. By definition of $\epsilon$, this number is larger than $\epsilon + 2$. This proves that there are at least $\epsilon + 2$ bins $B_k$ such that $m_k \in [m_f, M + \alpha L\sqrt{d}/K]$, so $m_{f-\epsilon-2} \leq M + \alpha L\sqrt{d}/K$. Therefore, $m_{\hat{f}} \leq M + \alpha L\sqrt{d}/K$ and $m_{\hat{f}+1} \leq M + \alpha L\sqrt{d}/K$ with probability larger than $1 - 2\exp\left(-\frac{2c_{p,d}^2 N}{K^2}\right)$.

Similarly, we can show that $m_{\hat{f}} \geq M - \alpha L\sqrt{d}/K$ and $m_{\hat{f}+1} \geq M - \alpha L\sqrt{d}/K$ with probability larger than $1 - 2\exp\left(-\frac{2c_{p,d}^2 N}{K^2}\right)$.

### D.18 Proof of Lemma 22

Non-zero terms in $R_T^{(d)}$ correspond to pairs of arms $(i, j)$ such that $i$ is pulled by $\phi^d$ but not by $\phi^*$, and $j$ is pulled by $\phi^*$ but not by $\phi^d$. If an arm $i$ is pulled by $\phi^d$, it belongs to a bin $k$ such that $m_k \geq m_{\hat{f}+1}$. On the event $\mathcal{E}_a$, $m_{\hat{f}+1} \geq M - \alpha\sqrt{d}L/K$. Using Lemma 17, we find that

$$m(a_i) \geq M - 2\alpha\sqrt{d}L/K.$$

On the other hand, if $i$ is not pulled by $\phi^*$, it must be such that $m(a_i) \leq \widehat{M}$. On the event $\mathcal{E}_a$, this implies that $m(a_i) \leq M + \sqrt{d}L/K$. Since there are at most $\frac{4\alpha\sqrt{d}LQN}{K}$ arms in $[M - 2\alpha\sqrt{d}L/K, M + \sqrt{d}L/K]$ on the event $\mathcal{E}_a$, there are at most $\frac{4\alpha\sqrt{d}LQN}{K}$ arms that are selected by $\phi^d$ and not by $\phi^*$, and thus at most $\frac{4\alpha\sqrt{d}LQN}{K}$ non-zero terms in $R_T^{(d)}$.

Similarly to the one-dimensional case, the cost of pulling an arm $i$ selected by $\phi^d$ but not by $\phi^*$, instead of an arm $j$ selected by $\phi^*$ but not by $\phi^d$, is bounded by $2|\widehat{M} - m_{\hat{f}+1}| \vee 2\sqrt{d}L/K \leq 2\alpha\sqrt{d}L/K$. To conclude, on the event $\mathcal{E}_a$ there are at most $\frac{4\alpha\sqrt{d}LQN}{K}$ non-zero terms in $R_T^{(d)}$, and each of them is bounded by $2\alpha\sqrt{d}L/K$. Thus,

$$R_T^{(d)} \leq \frac{8\alpha^2 dQL^2 N}{K^2}.$$

### D.19 Proof of Lemma 23

Note that for $k \in \{1, ..., K\}$, $B_k$ is a Borel set of measure $1/K^d$. Applying Bernstein's inequality, we find that for all $t > 0$,

$$\mathbb{P}\left(\left|N_k - \frac{N}{K^d}\right| \geq t\right) \leq 2e^{\frac{-t^2}{2K^{-d}N + 2t/3}}.$$

Choosing $t = K^{-d}N/2$, we find that

$$\mathbb{P}\left(\left|N_k - \frac{N}{K^d}\right| \geq \frac{N}{2K^d}\right) \leq 2e^{-\frac{N}{10K^d}}.$$

A union bound for $k = 1, ..., K^d$ yields the result.

### D.20 Proof of Lemma 26

By Lemma 19, on the event $\mathcal{E}_a$, $m_{\hat{f}+1} \in [M - \alpha\sqrt{d}L/K, M + \alpha\sqrt{d}L/K]$. We group bins with mean rewards lower than $m_{\hat{f}+1}$ into the following subsets.

Let $\mathcal{S}_0 = \left\{k : (M - m_k) \in [-\alpha\sqrt{d}L/K, 2\alpha\sqrt{d}L/K]\right\}$, and for $n \geq 1$ define $\mathcal{S}_n = \left\{k : (M - m_k) \in [2^n\alpha\sqrt{d}L/K, 2^{n+1}\alpha\sqrt{d}L/K]\right\}$. Note that for $n \geq \log_2(K/(\alpha\sqrt{d}L))$, $\mathcal{S}_n$ is empty since $m$ is bounded by 1.

Using Lemma 17, we note that for all $l \in \mathcal{S}_0$ and all $a \in B_l$, $|m(a) - M| \leq 4\alpha\sqrt{d}L/K$. Using Assumption 3, we conclude that $|\mathcal{S}_0| \leq 4\alpha\sqrt{d}LQK^{d-1}$. On $\mathcal{E}_a$, there are at most $1.5N/K^d$ arms in each bin, so the number of arms in bins in $\mathcal{S}_0$ is at most $6\alpha\sqrt{d}LQN/K$. Moreover for all $l \in \mathcal{S}_0$ and all $a_i \in B_l$, $(M - m(a_i)) \leq 4\alpha\sqrt{d}L/K$. Thus, the arms pulled from bins in $\mathcal{S}_0$ contributes to $R_{subopt}$ by at most $24\alpha^2 dQL^2 N/K^2$.

Similarly, for all $n \geq 1$, all $l \in \mathcal{S}_n$ and all $a \in B_l$, $|m(a) - M| \leq 2^{n+1}\alpha\sqrt{d}L/K$. Using Assumption 3, we conclude that $|\mathcal{S}_n| \leq 2^{n+1}\alpha\sqrt{d}LQK^{d-1}$. Moreover, by definition of $\hat{f}$, there exists a bin $B_k$ with $m_k \geq m_{\hat{f}+1}$ such that $n_k(T) < N_k$. Since $\Delta_{k,l} \geq 2^n\alpha\sqrt{d}L/K - \alpha\sqrt{d}L/K \geq 2^{n-1}\alpha\sqrt{d}L/K$ for all $l \in \mathcal{S}_n$, we use Lemma 25 and find that

$$n_l(T) \leq \frac{3\log(T/\delta)K^2}{2^{2n-2}\alpha^2 L^2 d}.$$

Thus,

$$
\begin{aligned}
R_{subopt} &\leq \frac{24\alpha^2 dL^2 QN}{K^2} + \sum_{n=1}^{\log_2(K/\alpha\sqrt{d}L)} 2^{n+2}\alpha\sqrt{d}QLK^{d-1} \times \frac{3\log(T/\delta)K^2}{2^{2n-2}\alpha^2 L^2 d} \times \frac{2^{n+1}\alpha\sqrt{d}L}{K} \\
&\leq \frac{24\alpha^2 dL^2 QN}{K^2} + 96QK^d \log(T/\delta)\log_2(K/\alpha\sqrt{d}L).
\end{aligned}
$$

## D.21 Proof of Lemma 27

Using the notations and results established along the proof of Lemma 26, we find that

$$
\begin{aligned}
\sum_{n=0}^{\log_2(K/\alpha\sqrt{d}L)} \sum_{B_k \in \mathcal{S}_n} n_k(T) &\leq 6\alpha\sqrt{d}LQN/K + \sum_{n=1}^{\log_2(K/\alpha\sqrt{d}L)} 2^{n+2}\alpha\sqrt{d}QLK^{d-1} \times \frac{3\log(T/\delta)K^2}{2^{2n-2}\alpha^2 L^2 d} \\
&\leq 6\alpha\sqrt{d}LQN/K + \frac{24\log(T/\delta)K^{d+1}Q}{\alpha\sqrt{d}L}.
\end{aligned}
$$

## D.22 Proof of Lemma 28

As in the one-dimensional case, we use the following notations : for any two bins $B_h$ and $B_l$ such that $m_h \geq m_l$, define $N_{[h,l]} = \sum_{k=h}^{l} N_k$, and $n_{[h,l]}(T) = \sum_{k=h}^{l} n_k(T)$. We prove Lemma 28 by contradiction. We assume that there exists a bin $B_k$ such that $m_k \geq M + A\sqrt{d}L/K$ and $n_k(T) < N_k$ and define $h$ such that $m_h \in \arg\max_l\{m_l : m_l \leq M + A\sqrt{d}L/(2K)\}$. Then, the arguments used to prove Lemma 12 show that we necessarily have $N_{[h,\widehat{f}]} < n_{[h,\widehat{f}]}(T) + n_{[\hat{f}+1,K^d]}(T)$. We obtain a contradiction by proving that on $\mathcal{E}_a \cap \mathcal{E}_b \cap \{n_k(T) < N_k\}$,

$$
N_{[h,\widehat{f}]} - n_{[h,\widehat{f}]}(T) > n_{[\hat{f}+1,K^d]}(T).
$$

To obtain a lower bound on $N_{[h,\widehat{f}]} - n_{[h,\widehat{f}]}(T)$, we note that for all $l \in [h, \hat{f}]$, $\Delta_{k,l} \geq \Delta_{k,h} \geq A\sqrt{d}L/(2K)$. Using Lemma 9, we see that of the event $\mathcal{E}_a \cap \mathcal{E}_b \cap \{n_k(T) < N_k\}$

$$
n_l(T) \leq \frac{3\log(T/\delta)}{\Delta_{k,l}^2} \leq \frac{3\log(T/\delta)}{\left(A\sqrt{d}L/(2K)\right)^2} \leq \frac{12K^2\log(T/\delta)}{\left(A\sqrt{d}L\right)^2}.
$$

On the event $\mathcal{E}_b$, each bin contains at least $N/2K^d$ arms. Thus,

$$
N_h - n_h(T) \geq \frac{N}{2K^d} - \frac{12K^2\log(T/\delta)}{\left(A\sqrt{d}L\right)^2}.
$$

The following reasoning helps us obtain a lower bound on the number of bins $B_l$ for $l \in [h, \widehat{f}]$, denoted by $\mathcal{N}_{[h,\widehat{f}]}$. First, recall that on $\mathcal{E}_a$, $m_{\widehat{f}} \leq M + \alpha\sqrt{d}L/K$. Now, any arm $a$ such that $m(a) \in [M + 2\alpha\sqrt{d}L/K, M + A\sqrt{d}L/4K]$ belongs to a bin $B_l$ such that $m_l \in [M + \alpha\sqrt{d}L/K, M + A\sqrt{d}L/2K]$. By definition of $h$, this bin $B_l$ is such that $l \in [h, \widehat{f}]$.

Next, we use Lemma 18 to lower bound $\lambda\left(\{a : m(a) \in [M + 2\alpha\sqrt{d}L/K, M + A\sqrt{d}L/4K]\}\right)$. We have assumed that there exists a bin with mean reward larger than $M + A\sqrt{d}L/K$, so we necessarily have $A/K \leq 1$. Using Assumption 3 and Lemma 18, we find that for the constant $c_{p,d}$

appearing in Lemma 18

$$\lambda \left( \{a : m(a) \in [M + 2\alpha\sqrt{d}L/K, M + A\sqrt{d}L/4K]\} \right)$$
$$= \lambda \left( \{a : m(a) \in [M, M + A\sqrt{d}L/4K]\} \right) - \lambda \left( \{a : m(a) \in [M, M + 2\alpha\sqrt{d}L/K]\} \right)$$
$$\geq c_{p,d} \frac{A\sqrt{d}}{4K} - 2\alpha\sqrt{d}QL/K.$$

By definition of $A$, we have $A \geq 16\alpha QL/c_{p,d}$, and thus $c_{p,d}A\sqrt{d}/(4K) - 2\alpha\sqrt{d}QL/K \geq c_{p,d}A\sqrt{d}/(8K)$. Now, all arms in $\{a : m(a) \in [M + 2\alpha\sqrt{d}L/K, M + A\sqrt{d}L/4K]\}$ belongs to bins in $[h, \widehat{f}]$. Since each of those bins have volume $K^{-d}$, we find that $\mathcal{N}_{[h,\widehat{f}]} \geq c_{p,d} \frac{A\sqrt{d}}{8} K^{d-1}$, and

$$N_{[h,\widehat{f}]} - n_{[h,\widehat{f}]}(T) \geq \frac{A c_{p,d}\sqrt{d}K^{d-1}}{8} \left( \frac{N}{2K^d} - \frac{12K^2 \log(T/\delta)}{\left(A\sqrt{d}L\right)^2} \right).$$

To obtain an upper bound on $n_{[\hat{f}+1,K]}(T)$, we divide the bins $\hat{f} + 1, ..., K$ into subsets. Let $\widetilde{\mathcal{S}}_0 = \{l : M - m_l \in [-\alpha\sqrt{d}L/K, A\sqrt{d}L/K\}$, and for $n > 0$ let $\widetilde{\mathcal{S}}_n = \{l : M - m_l \in [A\sqrt{d}L/K \times 2^{n-1}, A\sqrt{d}L/K \times 2^n\}$. Since $m_{\hat{f}} \leq M + \alpha\sqrt{d}L/K$, we see that $\{\hat{f} + 1, ..., K\} \subset \underset{n\geq 0}{\cup} \widetilde{\mathcal{S}}_n$.

For all $l \in \widetilde{\mathcal{S}}_0$, $\Delta_{k,l} \geq (A - \alpha)\sqrt{d}L/K \geq 15A\sqrt{d}L/(16K)$ since $A > 16\alpha$. Using Lemma 17 and Assumption 3, we find that $|\widetilde{\mathcal{S}}_0| \leq 2A\sqrt{d}LQK^{d-1}$. Similarly, for all $n > 0$ and all $l \in \widetilde{\mathcal{S}}_n$, $\Delta_{k,l} \geq A\sqrt{d}L(1 + 2^{n-1})/K \geq A\sqrt{d}L2^{n-1}/K$, and $|\widetilde{\mathcal{S}}_n| \leq A\sqrt{d}QL2^{n+1}K^{d-1}$. Using Lemma 25, we find that on the event $\mathcal{E}_a \cap \mathcal{E}_b \cap \{n_k(T) < N_k\}$,

$$\begin{aligned} n_{[\hat{f}+1,K]}(T) &\leq \frac{768K^{d+1}Q\log(T/\delta)}{225\sqrt{d}AL} + \sum_{n\geq 1} AQ\sqrt{d}L2^{n+1}K^{d-1} \frac{3K^2 \log(T/\delta)}{A^2 dL2^{2n-2}} \\ &\leq \frac{28K^{d+1}Q\log(T/\delta)}{AL\sqrt{d}} \end{aligned}$$

Recall that we necessarily have $QL \geq 1$, and that $c_{p,d} \leq 1$. Thus, for the choice

$$A = \sqrt{\frac{472QK^{d+2} \log(T/\delta)}{Nc_{p,d}Ld}} \vee 16\alpha QL/c_{p,d},$$

we find that $N_{[h,\widehat{f}]} - n_{[h,\widehat{f}]}(T) > n_{[\hat{f}+1,K]}(T)$, which is impossible. We conclude that all bins $B_l$ with a mean reward larger than $M + A\sqrt{d}L/K$ have been emptied.