[Reviews · NeurIPS 2020]

Review 1

Summary and Contributions: This paper introduces the problem of finite continuum armed bandits with N arms in [0,1] and the constraint that each arm can be pulled at most once. Learning remains possible thanks to regularity of the reward function. For this problem, it provides an algorithm that is optimal (up to constants and polylog factors). Surprisingly, its regret bound is T^{1/3}, while the minimax bound is actually T^{1/2} in most bandits problem. Besides proving its upper bound, the paper thus also proves a lower bound.

Strengths: By adding a new constraint to the continuum armed bandit problem, this work shows that the reached regret is actually way smaller (but not the total reward). This result is rather surprising at first and very nice. The theoretical analysis is smooth and rather complete.

Weaknesses: Although the elements of understanding why we get a T^{1/3} bound instead of T^{1/2} are given in the paper, I think they might have been better highlighted. (I give my suggestions and how I felt when reading the paper in the additional feedback section). Slightly related to this, I think that the assumptions could be better justified when presented. (see additional feedback section as well). These are only minor weaknesses in my opinion, justifying my overall score.

Correctness: The proofs seem correct to me. I might have missed some minor bugs in them, but the major elements seem ok.

Clarity: I find the paper well written globally. But as mentioned in the weaknesses section, I think that some points might have been presented differently, and especially the reasons of why we have T^{1/3} and not T^{1/2}.

Relation to Prior Work: The relation to prior work is clearly discussed. Yet when reading the introduction (esp. the motivations of this model), I think it is necessary to explain the differences/complementarity with contextual bandits.

Reproducibility: Yes

Additional Feedback: I have read the authors' response and the other reviews. I had no personal concern about this work and thus maintain my score. ********************************************************* - While reading the paper at first, I thought for a long time that the regret T^{1/3} (which is mainly a consequence of remark 1) was more due to Assumption 3 than to the constraint of one pull per arm as explained by the authors. It was actually a combination of the two, as assumption 3 already allowed to improve from T^{2/3} to T^{1/2} in the continuum armed bandit (Auer et al., 2007). This is mentioned in the paper but only at page 7. I would personally prefer to see it just after introducing Assumption 3. Besides justifying this assumption, it would also suggest that it already improves the regret in the continuum armed bandit, but not enough to reach T^{1/3}. - How is assumption 2 local lipschitz ? This is not the definition of local lipschitz I know and taking the max with M-m(x) is actually weird. It would be great to have a discussion about this. - Still on assumption 2, I think that this work can be easily extended to any apha-Hölder reward function. It might be great to at least quickly mention it after assumption 2. ******* Minor remarks ********** - Line 83: the sentence about the work of Chakrabarti et al. is confusing. These are not the rewards that are drawn i.i.d. from some known distribution, but the "mean rewards". - In the for loop of the algo, (last - ), it seems that you actually wanted to define I_k as [k-1/K, k/K) \cap { a_1, ..., a_N } - there is a typo in the title on the CMT submission. This might cause trouble in the proceedings in case of acceptance Some typos I noticed: - line 80 "where" -> "were" - line 165 "maximas" -> "maxima" - line 174 "obtains" -> "obtained"


Review 2

Summary and Contributions: This work proposed a novel bandit study framework called finite continuum-armed bandits, where on top of the classical continuum-armed bandits setting, the authors added a constraint that each arm could be pulled at most once. Furthermore, they assumed that the total budget is a fixed proportion p \in (0,1) of the number arms. The authors proved a O(T^(1/3)) lower bound and designed a matching algorithm that achieves the optimal bound up to logarithmic factors. After rebuttal and discussion: I'm pretty satisfied with the authors answer regarding the dependency on p. It seems however that the assumption 1 is still questionable, I thus decide to set my final score to 6.

Strengths: This paper achieved better regret bounds under a rather novel setting (to the best of my knowledge) compared to the classical CAB setting, which is an interesting result.

Weaknesses: I have several questions however that I would like the authors to elaborate a bit more. First of all, could the authors elaborate a bit more on the motivation of this setting (I agree that it is theoretically interesting)? The motivating examples are not that clear to justify the assumption of fixing T as a fraction p of N. For me an unknown p would be more realistic (which is the case for the classical CAB or many-armed bandits). This is important to me since it seems that the gain of the bound comes from this p. If we take the extreme case where N -> \infty, we will reduce to the classical CAB case (as mentioned by the authors). This is equvalent to consider p -> 0 actually, and in that case, it appears to me that the current lower bound in the paper does not make any sense. So I would like to ask if the authors could give more intuition on how this p affects the analysis. Finally, is that possible to provide some light experimental illutrations? The algorithm looks implementable, and it could be interesting to see how it behaves in practice.

Correctness: The paper has pure theoretical contributions, the claims are correct.

Clarity: The paper is sound in terms of writting.

Relation to Prior Work: It is quite clear for me that how the new setting differs from the previous ones. However, it seems to me that the paper still lacks some discussion over some related work, in particular the line of research on X-armed bandits such as Bubeck et al. 2011, Slivkins 2011, Munos 2011, Bull 2015, Locatelli and Carpentier 2018, etc (the list is non-exhaustive). Indeed, in those papers, even though it is not explicitely defined, you usually do not sample one arm more than once. I would also suggest the authors to discuss a bit the assumptions they made compared to e.g. Kleinberg 2004, Minsker 2013, Locatelli and Carpentier 2018.

Reproducibility: No

Additional Feedback: The major minor comment: the title in CMT is wrong (missing 'n'). Minor comments: - Line 133: in the F-CAB setting than in the less constrained -> in the F-CAB setting than that in the less constrained... - Line 149: the cumulative rewards are lower than in the classical multi-armed -> are lower than those in the classical...


Review 3

Summary and Contributions: The paper introduces a continuum-armed bandit variant, where the player can pull discrete arms only once. Under some assumptions, the algorithm proposed for the new variant achieves O(T^1/3) regret. A matching lower bound is also provided.

Strengths: The new problem is interesting, and the theoretical work is valuable.

Weaknesses: Some assumptions seems unnatural, resulting in a questionable applicability of the problem.

Correctness: The paper is mostly sound.

Clarity: The paper is mostly clearly written.

Relation to Prior Work: It is sufficient.

Reproducibility: Yes

Additional Feedback: There are many applications where arms die (or degrade) after being pulled. Assumptions 2 and 3 are similar to the ones in [Auer et al. 2007] with \alpha=1 and \beta=1, which are on the easier side. The order of the regret shifting from T^2/3 of Kleinberg to T^1/2 is in fact due to \beta (\beta=1 being the easier problem). The problematic assumption is Assumption 1. While it seems an `innocent' one, it is making much more difficult for nature to choose a difficult instance. Nature can choose a function, and then Assumption 1 is smoothing it (in expectation). It is also difficult for me to imagine an application where this assumption would be natural. The paper places a strong focus on the Mth arm. Choosing the first M arms is imperative for the optimal policy, but for a slightly sub-optimal policy, including the top M-1 arms would result only in constant regret. Probably it would be possible to construct an easy problem where the Mth arm is difficult but the top M-1 are easy (much better than the rest), but this is not working because Assumption 1. In Algorithm 1, the alive intervals are defined as those the have at least two arms. I assume that this a incorrect, and an alive interval is one that has at least one arm that has not been pulled. This should be corrected or clarified. While the regret of the algorithm is stated as O(T^1/3), the constant C_{L,Q,p} hides the nature of the dependency on p. It is clear that for some values of p, the regret should at least get closer to O(T^1/2). Given the pivotal role of the constant, it is necessary to make it clear and probably have a discussion on it. Right now, it is difficult to figure out even the dependence on p. ------------ I have read the authors' feedback. I hope that the autors will indeed include the discussion on $p$ in the revised version (in case of acceptance). I am not convinced at all of their argument on Assumption 1.

[Author Response · NeurIPS 2020]

We thank the reviewers for their valuable insights and comments, and for pointing out corrections in the manuscript.
We will implement their advice and address all of their remarks in the final version of the manuscript.

One of the referees raised a concern about Assumption 1 (random uniform distribution of the arms). Random design is
a classical modelling in many different topics (matrix completion, random design regression, supervised classification).
In the bandit literature, the problem of contextual bandits, similar in some respects to the Finite Continuum-Armed
Bandit (FCAB) problem, has been studied in Perchet and Rigolet (2013) under the assumption that contexts be randomly
distributed in $[0, 1]^d$. We underline that the FCAB problem was originally motivated by a pair matching problem,
where one aims at discovering unobserved edges on a graph. A typical modelling in this setting is that each node is
characterised by a random feature $\xi_i$, and that given those features the probability that two nodes $i$ and $j$ are linked by
an edge is given by a function of $\xi_i$ and $\xi_j$. This is the case in the well known graphon model; other models of graphs
with random covariates have also been considered in Deshpande et al. (2018).

More restrictive, is the assumption of uniform distribution of the arms. Yet, our proofs can readily be extended to cover
any distribution with a positive density on $[0, 1]$. We chose to focus on the uniform distribution to avoid burdening our
proofs with additional technical details. As stated, our aim is to consider the problem of resource allocation with limited
budget, which is of practical importance and has not been addressed in the literature, and to expose as simply as possible
the differences and similarities between this problem and related problems (among which is the Continuum-Armed
Bandits (CAB)).

We thank the referees for raising a very interesting question regarding the dependency of our regret bound on $p$. We
can already address this question without modifying our proofs. In our paper, we show that in the FCAB, when the
budget $T$ is a fixed proportion $p$ of the number of arms $N$, the regret scales as $T^{1/3} \log(T)^{4/3}$. On the other hand, as $p$
decreases, we expect the problem to reduce to a CAB. In the limit where $p$ is sufficiently small, we therefore expect
the regret to scale as $\sqrt{T} \log(T)$. Our answer indicates that this is indeed the case, and that a smooth transition occurs
between those two settings as $p = T/N$ goes to 0. We present our results before providing a sketch of proof.

To highlight the dependency of $R_T$ on $T$, we consider regimes where $T = cN^\alpha$ for some $\alpha \in [0, 1]$ and $c \in (0, 1/2]$ (the
choice $c \leq 1/2$ reflects the fact that we are interested in settings where $T$ may be small compared to $N$, and is arbitrary).
Defining $\epsilon_N = (2 \log \log(N)/3 - \log(c))/\log(N)$, we obtain that for $\alpha \in (2/3 + \epsilon_N, 1]$, $R_T \leq CT^{1/(3\alpha)} \log(T)^{4/3}$
with large probability for some constant $C$ depending on $Q$, $L$ and $c$. This bound is obtained for the optimal
choice of the parameters $K = N^{1/3} \log(N)^{-2/3}$ and $\delta = N^{-4/3}$. Our lower bound on the regret matches this
rate up to poly-logarithmic factors. The setting considered in our paper corresponds to the case where $\alpha = 1$, and
$R_T = O(T^{1/3} \log(T)^{4/3})$. As expected, as $\alpha$ decreases, the regret increases. When $\alpha \to 2/3 + \epsilon_N$, the regret is of the
order $\sqrt{T} \log(T)$, and $K = \sqrt{T}/\log(T)$. We underline that these values correspond respectively to the regret and the
optimal value for $K$ in the CAB. This should come as no surprise, as $\alpha = 2/3 + \epsilon_N$ corresponds to a transition from a
setting where the finiteness is predominant, to a setting where the difficulty of the problem is that of a CAB.

To understand this transition, note that when $\alpha = 2/3 + \epsilon_N$, we have $T = N/K$. We recall that all intervals contain
approximately $N/K$ arms. Thus, when $\alpha > 2/3 + \epsilon_N$, $T > N/K$, the oracle strategy exhausts all arms in the
best interval, and it must select arms in other intervals, so the finiteness is a constraining issue. On the contrary, if
$\alpha \leq 2/3 + \epsilon_N$, no interval is ever exhausted by any strategy, and our problem becomes very close to the CAB. The
oracle strategy only selects arms from the interval with highest mean reward. The analysis of the problem becomes
much simpler, as results can be directly inferred from Auer et al. (2007) by noticing that Algorithm UCBF never
exhausts any intervals, and is therefore a variant of Algorithm UCBC. In this case, the optimal choice for the number
of intervals remains $K = \sqrt{T}/\log(T)$, and yields a regret bound $R_T = O(\sqrt{T} \log(T))$ (choosing as previously
$K = N^{1/3} \log(N)^{-2/3}$ would produce too many intervals, which we would not have time to explore). The following
table summarises these results for $T = cN^\alpha$.

| $\alpha$ | $0 \leq \alpha \leq 2/3 + \epsilon_N$ | $2/3 + \epsilon_N < \alpha \leq 1$ |
|---|---|---|
| Optimal $K$ | $K = \sqrt{T}/\log(T)$ | $K = N^{1/3} \log(N)^{-2/3}$ |
| Regret | $R_T = O(\sqrt{T} \log(T))$ | $R_T = O(T^{1/(3\alpha)} \log(T)^{4/3})$ |

**Sketch of proof**: Along the proof of Theorem 1 (at line 486 in the Appendix), we show that for $\delta = N^{-4/3}$ and the
optimal choice $K = N^{1/3} \log(N)^{-2/3}$, with large probability $R_T \leq CN^{1/3} \log(N)^{4/3}$ for some $C$ depending only on
$L$ and $Q$ when $K > p^{-1} \wedge (1-p)^{-1}$. This bound can be rephrased as $R_T \leq CT^{1/(3\alpha)} \log(T)^{4/3}$, where $C$ depends on
$L$, $Q$ and $c$. We assumed that $T \leq N/2$, so $p \leq 1 - p$, and the condition $K > p^{-1} \wedge (1-p)^{-1}$ is met if $\alpha > 2/3 + \epsilon_N$.
On the other hand, Theorem 2 shows that with positive probability, $R_T \geq 0.01T^{1/3}p^{-1/3} = 0.01T^{1/(3\alpha)}c^{-1/(3\alpha)}$
when $N \geq N_{L,p}$, where $N_{L,p}$ is defined at line 504 in the Appendix. This condition is met when $N$ is larger than some
absolute constant, and $\alpha \geq 2/3 + C/\log(N)$ for some constant $C$ depending on $L$.

[Meta-Review · NeurIPS 2020]

The reviewers consider the submission as relatively significant. The post-rebuttal discussion was mainly on whether Assumption 1 is not too limiting/strong/unnatural as the random selection of the arms comes after the nature selects the payoff distribution. Therefore while the current setting with Assumption 1 seems artificial, one reviewer suggested that a more natural setting would be to just consider deterministic a_i of the form a_i = i/N (and actually any fixed a_i as soon as there is a separation gap between each a_i). thought that the results could be easily extended to this case, without carefully checking this. The authors are also the requested to include the discussion on $p$. NOTE FROM PROGRAM CHAIRS: Broader Impact statements are required in all papers submitted to NeurIPS 2020. Although this work is largely theoretical, one reviewer notes: "Although the authors claim that the societal impact section is not applicable, the motivating example, in particular the allocation of scarce resources, does seem to raise potential ethical questions: how shall we allocate scarce resource such as ICU during for example the current COVID-19 pandemic is clearly something to be taken seriously. ...it may not be that appropriate to use such examples without further discussing the potential ethical concerns."